# Difference Predictive Coding for Training Spiking Neural Networks

**Ville Karlsson, Nicklas Fianda & Joni-Kristian Kämäräinen**
Tampere University
Tampere, Finland
{ville.karlsson, nicklas.fianda}@tuni.fi
joni-kristian.kamarainen@tuni.fi

## Abstract

Predictive coding networks (PCNs) offer a local-learning alternative to backpropagation in which layers communicate residual errors, aligning well with biological computation and neuromorphic hardware. In this work we introduce *Difference Predictive Coding* (DiffPC), a spike-native PC formulation for spiking neural networks. DiffPC replaces dense floating-point messages with sparse ternary spikes, provides spike-compatible target and error updates, and employs adaptive threshold schedules for event-driven operation. We validate DiffPC on fully connected and convolutional architectures, demonstrating competitive performance on MNIST (99.3%) and Fashion-MNIST (89.6%), and outperforming a backpropagation baseline on CIFAR-10. Crucially, this performance is achieved with high communication sparsity, reducing data movement by over two orders of magnitude compared to standard predictive coding. DiffPC thus establishes a faithful, hardware-aligned framework for communication-efficient training on neuromorphic platforms.

## 1 Introduction

The error backpropagation algorithm has been fundamental to the success of deep learning, yet its core mechanisms are widely considered biologically implausible Salvatori et al. (2023). Key limitations include the requirement for global error signals—where synaptic updates depend on information transmitted across multiple layers, far beyond locally available signals—and the reliance on continuous-valued communication and gradients, in contrast to the brain's use of discrete, event-driven signals N'dri et al. (2024). These discrepancies create a significant gap between conventional artificial neural networks (ANNs) and biological neural systems.

Artificial intelligence does not need to replicate biology—airplanes do not flap their wings—but certain biological properties are worth emulating. Examples include the brain's energy efficiency and its ability to perform robust and adaptive computation with sparse, noisy, and low-precision signals N'dri et al. (2024). These observations align with the development of neuromorphic systems, which address the limitations of conventional von Neumann architectures by co-locating memory and computation to reduce data movement Al Abdul Wahid et al. (2024), thereby enabling substantially lower energy consumption—a key biological property worth emulating. Within this hardware setting, Spiking Neural Networks (SNNs) provide a natural computational model: information is represented not by continuous activations but by discrete spikes, as observed in the brain Olshausen & Field (1996; 2004). In SNNs, neurons communicate through sparse, asynchronous events rather than dense, synchronous updates Mainen & Sejnowski (1995); Cox et al. (2000), making them intrinsically well suited for low-power implementation.

One promising framework for training such systems is Predictive Coding (PC). Originating from neuroscience, PC theorizes that the brain functions as a prediction machine, continuously generating top-down predictions of sensory input while bottom-up signals convey only the residual prediction errors Rao & Ballard (1999); Friston (2005); Spratling (2017); Huang & Rao (2011); Keller & Mrsic-Flogel (2018). Importantly, PC relies on local learning rules, where synaptic updates depend only on the activity of adjacent pre- and post-synaptic neurons. This locality makes PC highly

compatible with the parallel and distributed organization of neuromorphic hardware N'dri et al. (2024); Salvatori et al. (2023).

However, despite its theoretical alignment with neuromorphic principles, the standard formulation of PC faces practical computational challenges. To infer neural activities, PC networks typically perform an iterative settling process, requiring multiple forward and backward passes of information to converge for a single input. In standard implementations, this process relies on dense, floating-point message passing, resulting in a computational overhead that notably exceeds that of backpropagation Rosenbaum (2022). This reliance on dense, continuous communication during the iterative phase can offset the efficiency gains sought by deploying SNNs on event-driven hardware. Therefore, while PC offers a solution to the global transport problem of backpropagation, its standard formulation does not fully exploit the sparsity and efficiency of neuromorphic substrates.

Combining SNNs with PC-based training is a natural research direction to address these issues Lan et al. (2022); Wacongne et al. (2012); Boerlin et al. (2013); Ororbia (2023). In this work, we introduce *Difference Predictive Coding* (DiffPC), an algorithm that reformulates the predictive coding framework for native implementation in SNNs. DiffPC seeks to address the communication overhead of standard PC by replacing dense, floating-point message passing with sparse, ternary spike-based communication. By employing spike-compatible state update rules and adaptive threshold schedules, DiffPC ensures that computation and message passing are event-driven, occurring when necessary to correct prediction errors. Our results indicate that DiffPC achieves accuracy matching or exceeding that of standard predictive coding networks (PCNs) and backpropagation trained models on benchmark datasets, while reducing the number of transmitted bits by more than two orders of magnitude compared to standard PC baselines.

- We propose *Difference Predictive Coding* (DiffPC), a spike-native framework based on novel update rules that transmit incremental state updates via sparse ternary spikes rather than broadcasting the full state, resulting in reduced communication costs.
- We introduce an adaptive threshold scheduling mechanism that enables the discrete spiking network to closely approximate the dynamics of standard continuous predictive coding with fewer timesteps.
- We empirically validate DiffPC on fully connected and convolutional architectures, demonstrating that it matches the accuracy of standard predictive coding and matches or exceeds that of Backpropagation on MNIST, Fashion-MNIST, and CIFAR-10, while reducing the number of transmitted bits by more than two orders of magnitude compared to standard predictive coding baselines.

## 2 RELATED WORKS

**Spiking neural networks.** SNNs compute with discrete events and update their state only upon spike arrivals, yielding sparse, asynchronous processing that maps well to neuromorphic substrates and modern accelerators Pfeiffer & Pfeil (2018); Tavanaei et al. (2019). This event-driven operation provides a natural match to the parallel, low-power architecture of neuromorphic hardware such as TrueNorth Akopyan et al. (2015), Loihi Davies et al. (2018), Loihi 2 Intel Corporation (2021b), and SpiNNaker Furber et al. (2014). Beyond neuromorphic-vision benchmarks, deep SNNs now achieve competitive accuracy on static datasets when equipped with convolutional backbones and carefully engineered neuron and normalization layers Hu et al. (2024).

Several software frameworks have been developed to simulate and train SNNs, including Brian2 Stimberg et al. (2019), NEST Gewaltig & Diesmann (2007), SpikingJelly Fang et al. (2023), and LAVA Intel Corporation (2021a). In this work we use LAVA to verify that our methods are compatible with the Intel hardware chip Loihi 2 Intel Corporation (2021b), but also provide a PyTorch implementation for easy verification and faster runtimes.

**SNN training.** A key challenge in SNN learning is that spike generation is non-differentiable, preventing direct application of backpropagation. Contemporary approaches can be divided into three main families, each with distinct accuracy, latency, and efficiency trade-offs Hu et al. (2024).

*(i) ANN→SNN conversion.* In this approach, a ReLU ANN is trained with backpropagation and mapped to an SNN under a rate or latency coding assumption Cao et al. (2015). Practical pipelines reduce activation–rate mismatch via weight and threshold normalization Diehl et al. (2015) or reset-

by-subtraction Rueckauer et al. (2017), and further tighten equivalence using quantization mapping Hu et al. (2023), clip-floor-shift activation Bu et al. (2023), and post-training parameter calibration Li et al. (2024). Conversion scales well and can match ANN accuracy with few timesteps, but training remains off-chip with dense floating-point communication, and residual conversion error or latency may erode energy benefits.

*(ii) Direct surrogate-gradient training.* By treating SNNs as recurrent systems unrolled over time, the zero derivative of the spike function is replaced with a smooth surrogate, enabling backpropagation through time (BPTT) Wu et al. (2018); Neftci et al. (2019). Representative methods include temporal-loss formulations (SpikeProp) Bohte et al. (2002) and time-based error reassignment (SLAYER) Shrestha & Orchard (2018). Later advances improved optimization and representation by learning neuron dynamics such as time constants Fang et al. (2021). Significant progress was also made in normalizing membrane dynamics across time via threshold-dependent scaling (tdBN) Zheng et al. (2021), time-varying parameter decoupling (BNTT) Kim & Panda (2021), input rescaling for uniform temporal distributions (TEBN) Duan et al. (2022), or direct membrane potential regulation (MPBN) Guo et al. (2023). Finally, other methods mitigate surrogate mismatch via gradient re-weighting (TET) Deng et al. (2022), information maximization objectives (IM-Loss) Guo et al. (2022), or learnable surrogate shapes Lian et al. (2023). These methods reach state-of-the-art accuracy with $\mathcal{O}(1-8)$ timesteps, but they still rely on global backward signals and dense communication, which can limit viability of on-chip training.

*(iii) Local plasticity.* Purely local rules, such as spike-timing-dependent plasticity (STDP) and reward-modulated variants, are aligned with both biology and hardware constraints. However, they typically require auxiliary classifiers and tend to underperform on complex tasks Diehl & Cook (2015); Kheradpisheh et al. (2018); Ororbia (2023).

Overall, these approaches highlight a central trade-off in SNN training: methods that achieve the highest accuracy typically rely on dense, non-local signals or off-chip training, while methods that are fully local and spiking have yet to consistently match this performance on complex tasks.

**Predictive coding.** Predictive coding (PC) has recently emerged as an alternative to backpropagation that is both biologically motivated and compatible with event-driven computation. In PC, each layer generates predictions of activity in the layer below, while only the residual prediction errors are communicated forward Rao & Ballard (1999); Friston (2005); Spratling (2017); Huang & Rao (2011). PC has been developed into a computational framework with formal links to backpropagation and variational inference Millidge et al. (2021); Rao & Ballard (1999). A central advantage is its use of local learning rules: synaptic updates depend only on the activity of adjacent pre- and post-synaptic neurons, making the framework well suited for distributed neuromorphic implementation N'dri et al. (2024); Salvatori et al. (2023).

Several works have sought to integrate PC with SNNs Lan et al. (2022); Ororbia (2023); Lee et al. (2024). The PC-SNN algorithm formulates predictive coding in time-to-first-spike (TTFS) encoding, where each neuron spikes at most once Lan et al. (2022). This achieves unmatched energy efficiency in terms of spikes, but runtime scales exponentially with input precision ($2^B$ timesteps for $B$-bit input) and must be predefined due to the single-spike restriction. Additionally, their training schema remains reliant on transmission of dense floating point numbers and is done on GPU. Recent work toward creating purely spiking predictive coding frameworks Ororbia (2023); Lee et al. (2024) has made significant progress. However, to evaluate performance on discriminative benchmarks like MNIST, these frameworks adopt a hybrid approach. The spiking network is first trained in a purely unsupervised manner to learn representations. Subsequently, its weights are frozen, and a separate, non-spiking linear classifier is trained post-hoc on rate-coded activities extracted from the network's final layer. This reliance on an external, non-spiking component for the final classification step means the reported accuracies do not reflect the performance of an end-to-end spiking system, complicating a direct assessment of their utility for fully neuromorphic deployment.

**Relation to DiffPC.** Event-driven 'gradient-by-spikes' approaches approximate backpropagation by discretizing gradients into spikes Bohte et al. (2000); Cai et al. (2024), which enables training and inference using only spikes. We apply a similar approach in our proposed Difference Predictive Coding. Distinct from prior work, DiffPC integrates these concepts through three key mechanisms: (1) a spike-based message passing protocol that adapts sparse ternary communication specifically for predictive coding error propagation; (2) a difference-based update rule that triggers communication

only upon state changes to minimize redundancy; and (3) a cyclic threshold scheduler designed to accelerate the convergence of discrete spiking states toward continuous PCN targets.

# 3 BACKGROUND

## 3.1 SPIKING NEURAL NETWORKS

SNNs are a class of artificial neural networks that mimic the behavior of biological neurons more closely than conventional neural networks. Neurons in spiking neural networks communicate through discrete spikes, or action potentials, when their membrane potential $V$ reaches a certain activation threshold $T_0$. The output of the neuron $i$ is a function of the potential $s_i(V_i)$. The neuron model utilized in this work is based on the difference equation of the Integrate-and-Fire neuron model:

$$V_i(t+1) = V_i(t) - T_\theta s_i(V_i(t)) + \sum_j w_{ij} s_j(t), \quad V_i(0) = b_i \ , \tag{1}$$

where $V_i(t)$ is the integration variable (membrane potential), $T_\theta$ is the threshold, $b_i$ is the bias and $w_{ij}$ is the weight of the synapse connecting the input neuron $j$ to the neuron $i$. The spike activation function $s_i(t) \in \{-1, 0, 1\}$ is

$$s_i(V_i(t)) := \begin{cases} 1 & \text{if } V_i(t) \geq T_\theta, \\ -1 & \text{if } V_i(t) \leq -T_\theta, \\ 0 & \text{otherwise} \ . \end{cases} \tag{2}$$

Activation of the Integrate-and-Fire-neuron output causes a voltage drop (damping) in the difference equation (1).

## 3.2 PREDICTIVE CODING

In the context of neural networks, PC proposes that each layer generates predictions of the activity in the next layer. The next layer then computes the error between the forward propagated prediction and the target activity. This error signal is then propagated backward[1] to update both the targets and synaptic weights of the previous layer. The learning objective is to reduce the overall prediction error in the network. Unlike the separate forward and backward passes of conventional deep learning.

Predictive Coding Networks (PCNs) involve a bidirectional local flow of information; predictions of current targets in one direction and prediction errors in another (Figure 1). The target activity is updated using the received prediction error and the synaptic weights using the target activity. The updates are local, which is a substantial difference from the backpropagation algorithm, where updates depend on a single error signal calculated at the final output and propagated backward through the entire network.

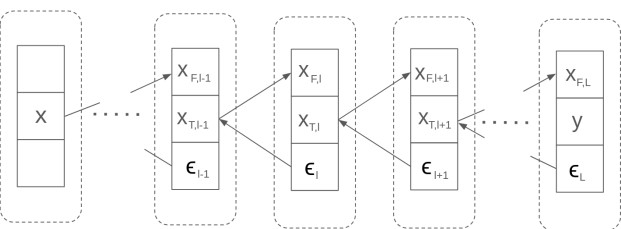

Figure 1: The structure of a multi-layer Predictive Coding Network (PCN). Each neuronal unit, bounded by the dashed line, consists of the target activity $\mathbf{x}_T$, prediction $\mathbf{x}_F$, and prediction error $\epsilon$. The arrows indicate the flow of information between layers $l = 0, 1, \ldots, L$, where the feedforward path carries the predictions from $\mathbf{x}_{T,l}$ to $\mathbf{x}_{F,l+1}$, and the feedback path conveys the prediction errors $\epsilon_l$ which are used to update $\mathbf{x}_{T,l-1}$. Computations and updates can be asynchronous.

---

[1]In the PCN literature the terms top/down are more common, but we opt forward/backward for consistency with the deep learning terminology.

We review the PC principles for a conventional Multi-Layer Perceptron network (MLP) of $L$ dense layers to the input $\mathbf{x}$ to output $\mathbf{y}$. The input and output layers are indexed as $l = 0$ and $l = L$, and between them are the hidden layers $l = 1$ to $L-1$. Predictive Coding (PC) makes use of each layer's current activation vectors $\mathbf{x}_{T,l}$ ('T' referring to 'Target'). In addition, the layers include vectors $\mathbf{x}_{F,l}$ (F: 'forward') that are the predictions generated by the previous layer $l - 1$. Ideally, the predictions and the targets should be the same.

**Energy function.** Layer $l$ predictions are calculated from the target activity of the layer $l - 1$

$$\mathbf{x}_{F,l} = \mathbf{W}_l \phi(\mathbf{x}_{T,l-1}) \ , \tag{3}$$

where $\mathbf{W}_l \in \mathbb{R}^{N_l \times N_{l-1}}$ are the weights, $N$ is the number of neurons, and $\phi(\cdot)$ is an activation function. The difference between the targets and predictions is fed back as the prediction error

$$\epsilon_l = \mathbf{x}_{T,l} - \mathbf{x}_{F,l} \ . \tag{4}$$

The errors from all layers are summed to compute the 'free-energy' of a network and PC operates by minimizing this free-energy function

$$\mathcal{F} = \sum_{l=1}^{L} \|\epsilon_l\|_2^2 = \sum_{l=1}^{L} (\mathbf{x}_{T,l} - \mathbf{x}_{F,l})^2 \ . \tag{5}$$

**Update steps.** The main difference between PC and the gradient descent error backpropagation is that PC alternates between the two updates, the target and weight updates, which are computed locally and can be asynchronous. The synaptic weight update is

$$\mathbf{W}_l^{(t+1)} = \mathbf{W}_l^{(t)} - \alpha \frac{\partial \mathcal{F}}{\partial \mathbf{W}_l} = \mathbf{W}_l^{(t)} + \alpha \epsilon_l \phi(\mathbf{x}_{T,l-1})^\top \ , \tag{6}$$

where $\alpha$ is the learning rate, and the prediction update is

$$\mathbf{x}_{T,l}^{(t+1)} = \mathbf{x}_{T,l}^{(t)} - \gamma \frac{\partial \mathcal{F}}{\partial \mathbf{x}_{T,l}} = \begin{cases} \mathbf{x}_{T,l}^{(t)} - \gamma \left( \epsilon_l - \mathbf{W}_{l+1}^\top \epsilon_{l+1} \odot \phi'(x_{T,l}) \right), & \text{for } l < L, \\ \mathbf{x}_{T,l}^{(t)} - \gamma \epsilon_l, & \text{for } l = L \end{cases} \ , \tag{7}$$

where $\gamma$ is the prediction learning rate and $\odot$ is the Hadamard product. Because updates only occur between adjacent layers, they must be performed iteratively to allow information to propagate throughout the entire network. See Appendix 7.1 for an intuitive explanation of predictive coding.

## 4 THE DIFFERENCE PREDICTIVE CODING ALGORITHM (DIFFPC)

In this section, we propose the Difference Predictive Coding (DiffPC) algorithm that implements the standard PC on SNNs. To ensure that our algorithm can be deployed on a neuromorphic chip, we used the instruction set of the Intel Loihi 2 neuromorphic chip. The algorithm was verified in the official simulator because access to the actual Loihi 2 hardware is limited to Intel partners. We provide the full simulator code and the pseudocode is given in Algorithm 1.

### 4.1 ALGORITHM OVERVIEW

To adapt the predictive coding framework from Section 3.2 for spiking neural networks, its floating-point computations and information transfer must be converted into discrete spikes (Algorithm 1). In this formulation, all information transmitted during the learning steps takes the form of a sequence of ternary values (-1, 0, 1).

In DiffPC, each unit maintains a *target state* $x_T$ and an *actual state* $x_A$. The target state $x_T$ represents the desired (target) activity, while the actual state $x_A$ attempts to follow $x_T$, aiming to minimize the difference between them. Their difference is reduced incrementally by steps proportional to an adaptive threshold $T_\theta$. The difference-based adjustments are communicated as spikes to subsequent layers, which integrate the incoming spiking information.

Two error variables, $e_T$ and $e_A$, function in the same way and represent the errors of the two states. $e_T$ is the target error and $e_A$ is the actual error that attempts to align with $e_T$. The error $e_A$ is adjusted

by steps proportional to $T_\theta$, which are then transmitted to subsequent layers as spikes. Since the threshold $T_\theta$ determines the step size of these updates, its schedule is critical for convergence. In the next section, we will introduce specific scheduling strategies.

**Feed Forward Initialization.** Before the iterative DiffPC process begins (as detailed in Algorithm 1), the network undergoes a feed forward initialization phase. In this phase, input spikes are propagated through the layers in a single pass without feedback error calculation. This rapidly establishes an initial estimate for the target activities $\mathbf{x}_T$ and predictions $\mathbf{x}_F$, reducing the number of subsequent iterative steps required for convergence. This phase effectively mimics a standard feedforward SNN inference step to prime the network state and can be implemented by utilizing graded spikes on the Loihi 2 chip. See section 7.2 of the appendix for a more detailed explanation of the algorithm.

---

**Algorithm 1** DiffPC Algorithm for Spiking Neural Network Training

---

**Input:** Spike signals $s_{\text{in}}$, $s_e$
**Process parameters:** threshold $T_\theta(t)$, learning rate $\gamma(t)$, weight lr $\alpha$
**Initialize:** $x_F$, $x_T$, $x_A$, $e_T$, $e_A$, $e_B$, $s_A$, $s_e$

1: **Feed Forward Initialization:** Propagate input to prime $x_F$
2: **for** each time step $t$ **do**
3: $\quad\quad s_{\text{in}}^l \leftarrow W^l s_A^{l-1}$ $\hfill \triangleright$ *Receive spike input*
4: $\quad\quad x_F^l \leftarrow x_F^l + s_{\text{in}}^l \cdot T_\theta(t-1)$ $\hfill \triangleright$ *Update forward prediction*
5: $\quad\quad$ **if** $\gamma^l(t) > 0$ **then**
6: $\quad\quad\quad\quad e_T^l \leftarrow x_T^l - x_F^l$ $\hfill \triangleright$ *Compute target error*
7: $\quad\quad x_T^l \leftarrow x_T^l + \gamma^l(t) \cdot (-e_T^l + (x_T^l > 0) \odot e_B^l)$ $\hfill \triangleright$ *Update Target Activity*
8: $\quad\quad$ **if** $\gamma^l(t) > 0$ **then**
9: $\quad\quad\quad\quad e_T^l \leftarrow x_T^l - x_F^l$ $\hfill \triangleright$ *Update target error*
10: $\quad\quad s_A^l \leftarrow \text{sign}(x_T^l - x_A^l) \odot (|x_T^l - x_A^l| > T_\theta(t))$ $\hfill \triangleright$ *Generate spikes*
11: $\quad\quad s_A^l \leftarrow s_A^l \odot (x_A^l + s_A^l \cdot T_\theta(t) > 0)$ $\hfill \triangleright$ *'Spiking ReLU'*
12: $\quad\quad x_A^l \leftarrow x_A^l + T_\theta(t) \cdot s_A^l$ $\hfill \triangleright$ *Update Actual Activity*
13: $\quad\quad$ s_out.send($s_A^l$) $\hfill \triangleright$ *Send state spikes*
14: $\quad\quad \triangleright$ *Propagate Error:* $\hfill \triangleleft$
15: $\quad\quad$ e_out.send($s_e^l$) $\hfill \triangleright$ *Send error spikes*
16: $\quad\quad$ e_in$^l \leftarrow (W^{l+1})^\top s_e^{l+1}$ $\hfill \triangleright$ *Receive error input*
17: $\quad\quad e_B^l \leftarrow e_B^l + T_\theta(t-1) \cdot$ e_in$^l$ $\hfill \triangleright$ *Accumulate incoming errors*
18: $\quad\quad s_e^l \leftarrow \text{sign}(e_T^l - e_A^l) \odot (|e_T^l - e_A^l| > T_\theta(t))$ $\hfill \triangleright$ *Generate error spikes*
19: $\quad\quad e_A^l \leftarrow e_A^l + T_\theta(t) \cdot s_e^l$ $\hfill \triangleright$ *Update actual error*
20: $W^l \leftarrow W^l + \alpha\, e_T^l\, \phi(x_T^{l-1})^\top$ $\hfill \triangleright$ **Update Weights**

---

## 4.2 $T_\theta$ AND $\gamma$ SCHEDULES

We present a *cyclic scheduler*, which allows for accurate approximation of the standard PC algorithm,

$$T_\theta(t) = \frac{2^m}{2^{t \bmod n}}, \quad \gamma(t) = g(t \bmod n) \ , \tag{8}$$

where

$$g(x) = \begin{cases} \gamma, & \text{if } x = 0 \\ 0, & \text{otherwise} \end{cases} \ , \tag{9}$$

where $n, t \in \mathbb{N}^+$ denote the cycle length and timestep index, respectively, $m \in \mathbb{Z}$ sets the initial magnitude of the threshold via $T_\theta(0) = 2^m$, and $t \bmod n$ denotes the modulo operation. The set of $n$ steps starting with $\gamma(t) = \gamma$ is referred to as $\gamma$-cycle and a set of timesteps during which we train the network with a single input and output pair is called an iteration. A single iteration therefore consists of multiple $\gamma$-cycles.

To theoretically motivate the precision of this approach, we establish the following bound on the quantization error demonstrating that, in the absence of error, the spiking states exponentially converge to their floating-point targets within a cycle.

**Theorem 4.1.** *Suppose that the target activity for layer l, denoted as $x_T^l$, satisfies $|x_T^l - x_A^l| < 2^{m+1}$ and $x_T^l > 0$. Then, after one $\gamma$-cycle of $n$ timesteps of a cyclic scheduler, the difference between the target activity and the actual activity $|x_T^l - x_A^l|$ is less than $2^{m+1-n}$.*

*Proof.* See Appendix 7.3. □

Though this method is able to attain great precision in the approximation of standard PC when we set $n$ large, it comes at the cost of extra timesteps and spikes. In practice, we observe that as the PC network converges, the changes in $e_T$ and $x_T$ become smaller. Thus we should also scale the cyclic scheduler to be smaller as the network converges allowing us to use smaller $n$ and still attain high accuracies. This is the motivation behind the cyclic decay scheduler defined as

$$T_\theta(t) = d(t \bmod T)\frac{2^m}{2^{t \bmod n}}, \quad \gamma(t) = g(t \mod n) \ , \tag{10}$$

where $d(t)$ is a decreasing function. We set

$$d(t) = \left(1 - (1-a)\frac{t}{T}\right) \ , \tag{11}$$

where $a \in (0, 1]$ such that $d(t \bmod T) \in (0, 1]$ and $T$ is the length of the iteration. In addition to the cyclic decay scheduler, we introduce the *constant decay scheduler*, defined as

$$T_\theta(t) = d(t \bmod T)\, c, \quad \gamma(t) = g(t \bmod n), \tag{12}$$

where $c \in \mathbf{R}^+$. This schedule maintains a fixed threshold $c$ that is scaled by the decay function $d(t)$, thereby reducing the update size over the course of an iteration.

## 5 EXPERIMENTS

### 5.1 DATA AND SETTINGS

We evaluate our method using Multi-Layer Perceptron (MLP) architectures as MLPs provide a clean and well-studied baseline for predictive coding, and we extend our evaluation to Convolutional Neural Networks (CNNs) on the CIFAR-10 dataset to demonstrate the method's applicability to convolutional networks. For empirical validation, we use the MNIST, Fashion-MNIST, and CIFAR-10 benchmarks. MNIST and Fashion-MNIST comprise 60,000 training and 10,000 test grayscale images of size $28 \times 28$ across 10 classes. CIFAR-10 consists of 50,000 training and 10,000 test color images of size $32 \times 32$ across 10 classes. We train fully connected networks with one or two hidden layers for the simpler tasks, utilizing dropout. For CIFAR-10, we utilize a convolutional architecture, consisting of two convolutional layers with $5 \times 5$ kernels and stride 2 (with 10 and 5 filters respectively), followed by three fully connected layers. All models are optimized using AdamW. Data augmentation includes random translation jitter for MNIST and random horizontal flips for CIFAR-10. We assess performance by test-set classification accuracy and by spike efficiency, quantified as the average number of activity and error spikes per neuron per sample during training. Our CIFAR-10 implementation is based on code from Rosenbaum (2022); Millidge et al. (2020); Whittington & Bogacz (2017).

**Selected baselines** For comparison, we focus on Convolutional and MLP networks and include both conventional floating-point and spiking implementations. In addition, we report results from spiking networks trained with alternative learning rules beyond predictive coding, providing context on how our approach compares to state-of-the-art non-PC methods

## 5.2 RESULTS

**Classification accuracy** On MNIST, DiffPC achieves high accuracy that matches previously reported results for non-convolutional spike-based methods. For example, DiffPC-L attains 99.3% accuracy and DiffPC-S reaches 98.2%, placing them on par with or above several recent SNN models, as shown in Table 1. On Fashion-MNIST, which presents a greater challenge, DiffPC also achieves competitive accuracy (Table 2). Finally, on CIFAR-10, DiffPC demonstrates effective scaling to convolutional architectures; DiffPC-Long achieves 65.6% accuracy, surpassing the standard backpropagation baseline of 63.5%, while DiffPC-Efficient reaches 63.3% (Table 4).

**Communication efficiency** Communication efficiency provides further insight into the advantages of DiffPC. On modern hardware, the energy cost of moving data is often comparable to, and in many workloads higher than, the cost of arithmetic operations Horowitz (2014); Lian et al. (2023). Because memory access and interconnect traffic can be orders of magnitude more energy-intensive than a multiply–accumulate, the number of floating-point values transmitted during training and inference is a key proxy for communication energy. In addition, spiking implementations require computation to unfold in discrete timesteps, and the number of timesteps needed for convergence strongly predicts runtime performance Li et al. (2023).

Table 3 reports the average number of bits transmitted per neuron during error propagation and the corresponding timestep counts on the MNIST task. Standard backpropagation transmits 32 bits per neuron in a single timestep, while predictive coding (PC-SE) requires 960 bits across 15 timesteps. SNN-based predictive coding (PC-SNN) is similar as they use floating point numbers during the training stage of their network. In contrast, DiffPC achieves orders-of-magnitude improvements: DiffPC-L transmits only 0.18 bits (0.09 spikes) per neuron on average across 120 timesteps, and DiffPC-S reduces this further to 0.08 bits (0.04 spikes) per neuron across 75 timesteps. These results demonstrate that DiffPC combines competitive accuracy with substantially improved efficiency in terms of communication.

We observe similar trends on the CIFAR-10 dataset, as detailed in Table 4. Here, we compare two configurations: DiffPC-Long, which utilizes a scheduler cycle length of $n = 16$, and DiffPC-Efficient, which employs a shorter cycle length of $n = 12$. Both configurations run for 15 cycles per sample. While the convolutional architecture utilizes higher-fidelity error messaging compared to the MLP used for MNIST, DiffPC retains a substantial efficiency advantage. DiffPC-Long requires only 1.9 bits per neuron, and DiffPC-Efficient further reduces this to 0.7 bits. Although these values are higher than those for MNIST, they remain significantly lower than the 32 bits required by backpropagation or the 960 bits used by standard PC, demonstrating that the communication sparsity of DiffPC scales effectively to convolutional networks.

Table 1: Comparison of the Test Accuracy of Different SNN and PC Models on the MNIST dataset. (FC denotes fully connected layers)

| Method | Network Architecture | Acc. (%) |
|---|---|---|
| Backpropagation | 784FC-1024FC-512FC-10FC | 99.3 |
| PC-SE (Standard PC) (Pinchetti et al. (2024)) | 784FC-128FC-128FC-128FC-10FC | 98.3 |
| STiDi-BP (Mirsadeghi et al. (2021)) | 40C5-P2-1000FC-10FC | 99.2 |
| SSTDP (Liu et al. (2021)) | 784FC-300FC-10FC | 98.1 |
| PC-SNN (Lan et al. (2022)) | 784FC-200FC-10FC | 98.1 |
| SRC-RNN (De Geeter et al. (2024)) | 784FC-512FC-512FC-512FC-10FC | 98.4 |
| FastSNN (Taylor et al. (2022)) | 784FC-1000FC-10FC | 97.9 |
| FastSNN (Taylor et al. (2022)) | 32C5-P2-64C5-P2-1000FC-10FC | 99.3 |
| **DiffPC-L (Ours)** | 784FC-1024FC-512FC-10FC | 99.3 |
| **DiffPC-S (Ours)** | 784FC-400FC-10FC | 98.3 |

Table 2: Comparison of the Test Accuracy of models on the Fashion-MNIST dataset.

| Method | Network Architecture | Acc. (%) |
|---|---|---|
| FastSNN (Taylor et al. (2022)) | 784FC-1000FC-10FC | 89.1 |
| FastSNN (Taylor et al. (2022)) | 32C5-P2-64C5-P2-1000FC-10FC | 90.6 |
| SRC-RNN (De Geeter et al. (2024)) | 784FC-512FC-512FC-512FC-512FC-512FC-10FC | 88.5 |
| **DiffPC-M (Ours)** | 784FC-1000FC-10FC | 89.6 |
| **DiffPC-S (Ours)** | 784FC-400FC-10FC | 89.2 |

Table 3: Average bits transferred during the error propagation stage of different models, along with the timesteps used on the MNIST task. (fp: floating-point operations; sp: spikes)

| Method | Ops | Network Architecture | Bits/N | Timesteps |
|---|---|---|---|---|
| Backpropagation | fp | 784FC-1024FC-512FC-10FC | 32 | 1 |
| PC-SE (Pinchetti et al. (2024)) | fp | 784FC-1024FC-512FC-10FC | 960 | 15 |
| PC-SNN (Lan et al. (2022)) | fp | 784FC-200FC-10FC | 960 | 15 |
| **DiffPC-L (Ours)** | sp | 784FC-1024FC-512FC-10FC | 0.18 | 120 |
| **DiffPC-S (Ours)** | sp | 784FC-400FC-10FC | 0.08 | 75 |

Table 4: Comparison of Test Accuracy and Efficiency (average bits per neuron during error propagation) on the CIFAR-10 dataset. (fp: floating-point; sp: spikes)

| Method | Ops | Network Architecture | Acc. (%) | Bits/N | Timesteps |
|---|---|---|---|---|---|
| Backpropagation | fp | 10C5S2-5C5S2-50FC-30FC-10FC | 63.5 | 32 | 1 |
| PC-SE (Pinchetti et al. (2024)) | fp | 10C5S2-5C5S2-50FC-30FC-10FC | 65.3 | 960 | 15 |
| **DiffPC-Long (Ours)** | sp | 10C5S2-5C5S2-50FC-30FC-10FC | 65.6 | 1.9 | 240 |
| **DiffPC-Efficient (Ours)** | sp | 10C5S2-5C5S2-50FC-30FC-10FC | 63.3 | 0.7 | 180 |

**Numerical precision –** We evaluated the numerical precision of *DiffPC* by comparing the final states $x_T$ obtained with standard predictive coding (PCN) and with our method. To quantify the approximation, we measured the absolute difference between the hidden-layer activations produced by the two algorithms.

In this experiment, we used a fixed multilayer perceptron (MLP) with architecture $128-200-10$. For each trial, we initialized the synaptic weights randomly but shared them between PCN and *DiffPC*, ensuring that differences arise solely from the approximation scheme. As inputs, we used i.i.d. random vectors sampled uniformly from $[-1, 1]^{128}$ and as output we similarly had i.i.d. random vectors sampled uniformly from $[-1, 1]^{10}$. We repeated the evaluation over 300 random weight initializations and inputs.

For each random trial, we computed the absolute difference between the final states $x_T$ of PCN and *DiffPC*. We observed that the error depends systematically on the scheduler parameters. Specifically, the error is larger when the number of approximation steps $n$ is small and the limit-decay value $a$ is large, and it decreases consistently as $n$ increases and $a$ decreases. This trend was robust across weight initializations and random inputs, showing that approximation precision can be tuned directly through scheduler parameters.

This constitutes a general test of numerical fidelity within the specified architecture and activation function for three reasons. First, it eliminates dataset-specific structure and labels, so the comparison probes only the update rules rather than task semantics. Second, by combining random bounded inputs with many random weights, it explores a wide region of the state space. Third, the use of shared weights across both algorithms isolates the approximation error from any modeling differences.

The results seen in Table 5 demonstrate that *DiffPC* provides a close approximation of standard PCN dynamics under random input conditions, confirming that the method faithfully reproduces PCN across a broad range of states for the given architecture.

Table 5: Mean absolute difference between DiffPC and standard PC, averaged over three seeds. The value after the $\pm$ symbol represents the sample standard deviation. Lower is better.

| $n$ \ $a$ | 1.0 | 0.5 | 0.25 | 0.1 |
|---|---|---|---|---|
| 3 | $0.1506 \pm 0.0051$ | $0.0627 \pm 0.0014$ | $0.0292 \pm 0.0014$ | $0.0168 \pm 0.0004$ |
| 4 | $0.0750 \pm 0.0018$ | $0.0312 \pm 0.0005$ | $0.0158 \pm 0.0005$ | $0.0106 \pm 0.0003$ |
| 5 | $0.0373 \pm 0.0006$ | $0.0169 \pm 0.0006$ | $0.0102 \pm 0.0004$ | $0.0083 \pm 0.0006$ |
| 6 | $0.0197 \pm 0.0005$ | $0.0107 \pm 0.0005$ | $0.0083 \pm 0.0005$ | $0.0075 \pm 0.0006$ |
| 7 | $0.0117 \pm 0.0002$ | $0.0085 \pm 0.0005$ | $0.0075 \pm 0.0006$ | $0.0072 \pm 0.0006$ |

## 6 CONCLUSION

In this work, we presented Difference Predictive Coding, a learning framework that reformulates standard predictive coding for native implementation in spiking neural networks. By replacing dense floating-point communication with sparse, event-driven ternary spikes, DiffPC addresses the data movement bottleneck that typically constrains on-chip training.

Our results on MNIST, Fashion-MNIST, and CIFAR-10 indicate that DiffPC approximates continuous predictive coding dynamics with high precision. Crucially, it achieves competitive classification accuracy while greatly reducing the number of transmitted bits compared to backpropagation and standard predictive coding baselines. These findings suggest that DiffPC offers a viable pathway for spiking based learning on neuromorphic systems.

Looking forward, a primary direction for future research in this domain is the evaluation of DiffPC on significantly deeper architectures. Recent advancements, such as $\mu$-PC Innocenti et al. (2025a), have shown that predictive coding can scale to deep ResNets when inference dynamics are stabilized. Since DiffPC is designed as a faithful discretization of PC, it is reasonable to hypothesize that these stabilization techniques would transfer to this spike-based framework. A valuable extension of this work would be to quantify the layer-wise deviation between DiffPC and continuous PC states in deep networks, establishing how the spike-communication window must scale to maintain approximation fidelity.

A complementary future direction concerns temporally correlated data. Recent work shows that when inputs evolve smoothly over time, PC inference can be warm-started from previous states, reducing inference iterations by half and substantially lowering the number of weight updates Zadeh-Jousdani et al. (2025). Prototype-based continual-learning methods similarly demonstrate that reducing update frequency yields large energy benefits on neuromorphic hardware such as Loihi 2 Hajizada et al. (2024; 2025). Since DiffPC is intrinsically event-driven—remaining silent during steady states and emitting spikes only on changes—leveraging temporal priors may further reduce both inference steps and plasticity operations. Quantifying these temporal-sparsity benefits represents another natural extension of the present work.

Finally, beyond algorithmic scalability and temporality, transitioning DiffPC from simulation to physical neuromorphic hardware remains a critical milestone. Deployment on platforms such as Intel Loihi 2 would allow for the assessment of the protocol under real-world hardware constraints and provide a rigorous verification of its potential energy efficiency advantages.

ACKNOWLEDGMENTS

This work was supported by the Academy of Finland (project no. 336357, PROFI 6 - TAU Imaging Research Platform).

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

# 7 APPENDIX

## 7.1 INTUITION OF PREDICTIVE CODING TRAINING

To clarify the mechanics of the learning process, we provide a step-by-step intuition of how a Predictive Coding Network (PCN) learns to classify inputs. Unlike Backpropagation, which calculates gradients of a loss function with respect to weights, PCN frames learning as an energy minimization problem involving local neuronal activities.

The training process for a single input-label pair $(\mathbf{x}, \mathbf{y})$ proceeds as follows:

1. **Prediction (Forward Pass):** The input $\mathbf{x}$ is clamped to the input layer. The network propagates activity forward layer-by-layer to generate a prediction at the output layer.

2. **Constraint (Clamping):** During training, the output layer is clamped to the correct label $\mathbf{y}$. This immediately creates a prediction error at the output layer (since the network's initial guess likely did not match $\mathbf{y}$).

3. **Relaxation (Bidirectional state/error flow):** This is the core of PC. The error at the output layer implies that the *penultimate* layer's activity was "wrong." This error flows backward, pulling the hidden layer neurons away from their original values toward states that *would have* produced the correct output. This happens iteratively across all layers. The network "relaxes" into a low-energy state where the activities are consistent with both the input and the correct label.

4. **Weight Update (Learning):** Once the neuron activities have shifted to this better configuration, the synaptic weights are updated locally. The update rule effectively says: "Change the weight so that next time, this input naturally produces this 'better' hidden activity."

## 7.2 Algorithm breakdown

**Update forward prediction –** Forward prediction $\mathbf{x}_F$ is updated using the incoming spike signal $\mathbf{s}_A$ from the previous layer,

$$\mathbf{x}_F^{(t+1)} \leftarrow \mathbf{x}_F^{(t)} + \mathbf{W}^l \mathbf{s}_A^{l-1} \cdot T_\theta(t-1) \ . \tag{13}$$

**Update Target Activity and Generate Spikes**

The core of the DiffPC algorithm lies in iteratively updating the target activity vector $\mathbf{x}_T$ for each layer and communicating changes via spikes. The process begins by adjusting $\mathbf{x}_T$ to minimize prediction error, followed by generating spikes based on the discrepancy between this new target and the layer's current state.

First, the target activity $\mathbf{x}_T$ is updated based on both the local prediction error $\mathbf{e}_T$ and the error propagated from the subsequent layer, which is accumulated in $\mathbf{e}_B$. This update, performed only when the learning rate $\gamma(t)$ is active, is defined as:

$$\mathbf{x}_T \leftarrow \mathbf{x}_T + \gamma(t) \cdot (-\mathbf{e}_T + (\mathbf{x}_T > 0) \odot \mathbf{e}_B) \ .$$

This rule closely mirrors the standard PC update in Equation 7. The term $-\mathbf{e}_T$ corrects for local prediction error, while the second term incorporates feedback from the next layer. The element-wise condition $(\mathbf{x}_T > 0)$ serves as the derivative of the ReLU activation function, ensuring that updates are only applied to active neurons. During training, the target activities of the input and output layers are clamped to the provided data and labels, respectively. During inference, only the input layer is clamped.

Next, the algorithm generates spikes to communicate the necessary adjustments for bringing the layer's *actual* state, $\mathbf{x}_A$, in line with the newly updated *target* state, $\mathbf{x}_T$. Instead of transmitting dense floating-point values, DiffPC sends sparse ternary spikes. A spike is generated only if the magnitude of the difference between the target and actual activity for a given neuron exceeds the adaptive threshold $T_\theta(t)$.

The activity spike vector $\mathbf{s}_A$ is computed as follows:

$$\mathbf{s}_A = \text{sign}(\mathbf{x}_T - \mathbf{x}_A) \odot (|\mathbf{x}_T - \mathbf{x}_A| > T_\theta(t)) \ , \tag{14}$$

where $\odot$ denotes the Hadamard product. The $\text{sign}(\cdot)$ function determines the spike's polarity (+1 or -1), while the comparison operator produces a binary mask, ensuring that spikes are only generated when the required update is significant. This event-driven mechanism ensures that communication is sparse, as spikes are only transmitted to correct meaningful deviations from the target state.

**Spiking ReLU –** To implement a non-linear transfer function similar to the Rectified Linear Unit (ReLU) in conventional neural networks, we propose a masking operation that effectively prevents the actual neural activity $\mathbf{x}_A$ from becoming negative.

$$\mathbf{s}_A^+ \leftarrow \mathbf{s}_A \odot (\mathbf{x}_A + \mathbf{s}_A \cdot T_\theta(t) > 0) \ . \tag{15}$$

The spiking ReLU ensures that only correction spikes in $\mathbf{s}_A$ maintaining $\mathbf{x}_A \geq 0$ are allowed, effectively implementing a ReLU-like activation function. We can also implement a clipped ReLU activation function in a similar manner by setting an additional constraint:

$$\mathbf{s}_A^+ \leftarrow \mathbf{s}_A \odot (1 > \mathbf{x}_A + \mathbf{s}_A \cdot T_\theta(t) > 0) \ . \tag{16}$$

**Update activation –** The spiking ReLU produces a simple update step to update the actual activity $\mathbf{x}_A$:

$$x_A \leftarrow x_A + T_\theta(t) \cdot \mathbf{s}_A^+ \ . \tag{17}$$

The activation update operation adjusts $\mathbf{x}_A$ towards $\mathbf{x}_T$.

**Error encoding in spikes –** The target update is the same as in the standard PC in Sec. 3.2.

The target error vector $\epsilon_l$ is computed using (4). Similarly to the original PC, this error term received from the following layer serves as a measure of how well the current forward prediction matches the target activity. However, unlike the standard PC we cannot directly use the update rule (7) since the error is encoded in the form of spikes. Instead, after computing the target error $e_T = \epsilon_l$, the DiffPC algorithm generates error spikes $\mathbf{s}_e$ based on the difference between $\mathbf{e}_T$ and the actual error $\mathbf{e}_A$,

$$\mathbf{s}_e = \text{sign}(\mathbf{e}_T - \mathbf{e}_A) \odot (|\mathbf{e}_T - \mathbf{e}_A| > T_\theta(t)) \quad . \tag{18}$$

The spikes $\mathbf{s}_e$ are then sent to the following layers as error signals.

**Accumulate incoming errors –** Errors from the next layers are integrated into the network using the accumulated error vector $e_B$. This vector represents the sum of the incoming error signals weighted by the threshold $T_\theta(t)$,

$$e_B \leftarrow e_B + T_\theta(t) \cdot \text{e\_in} \quad ,$$

where e\_in denotes the incoming error signals from the next layer. The accumulated error vector $e_B$ helps refine the target activity $x_T$ by incorporating feedback from different layers of the network.

The actual error $e_A$ is then updated using previously generated error spikes $s_e$,

$$e_A \leftarrow e_A + T_\theta(t) \cdot s_e \quad .$$

**Weight Update Mechanism –** The update rule for a single sample, derived from minimizing free energy, is computed as $\Delta W_{ij} \propto e_{T,i} \cdot \phi(x_{T,j})$, where $e_{T,i}$ is the post-synaptic error state and $x_{T,j}$ is the pre-synaptic activity state. This computation is compatible with neuromorphic hardware like Loihi 2, which supports fixed-precision multiplication and accumulation of local variables.

## 7.3 Proof of Convergence

Here we provide the proof for Theorem 4.1 regarding the convergence of the cyclic scheduler.

*Proof.* Consider the cyclic scheduler where $T_\theta(0) = 2^m$. At the first timestep $t = 0$, if $2^{m+1} > |x_T^l - x_A^l| > 2^m$, then $x_A^l$ is updated by $2^m$. Consequently, after the update, we have:

$$|x_T^l - x_A^l| < 2^m.$$

The same trivially holds if $|x_T^l - x_A^l| < 2^m$ already held on the first timestep. At the next timestep $t = 1$, with $T_\theta(1) = 2^{m-1}$, the difference $|x_T^l - x_A^l|$ can again be reduced by $2^{m-1}$ if it exceeds $2^{m-1}$. Repeating this process over $n$ timesteps, each reduction step halves the threshold compared to the previous timestep from which the result follows by induction. $\square$

## 7.4 Hyperparameter Analysis

To understand the impact of our key scheduler hyperparameters, the cycle length n and the decay factor a, we performed an extensive grid search. The results, visualized in Figure 2, reveal a clear trade-off between classification accuracy, communication cost (spikes), and runtime (timesteps).

Figure 2a shows that, with a fixed decay (a=1.0), increasing the cycle length n generally improves performance. On both datasets, the accuracy gains diminish as performance saturates for sufficiently large n. However, this accuracy gain comes at a direct cost. By definition, a larger n value increases the number of timesteps per iteration, which in turn increases both the total runtime and the number of spikes transmitted.

The role of the adaptive decay parameter a is to tune the precision of spike-based communication within a fixed number of timesteps. Figure 2b illustrates this trade-off effectively. For smaller cycle lengths, reducing a from 1.0 to smaller values provides a notable accuracy boost. This performance gain is achieved by allowing the adaptive threshold to decrease more over the iteration, which in turn generates more spikes to represent the error signals with higher fidelity. This can increase communication cost, but does not increase the runtime as the number of timesteps per iteration is fixed by n.

Crucially, the benefit of a smaller a diminishes as n increases. For large n, the performance is already high and stable, and varying a has little to no effect on the final accuracy. This suggests that a long

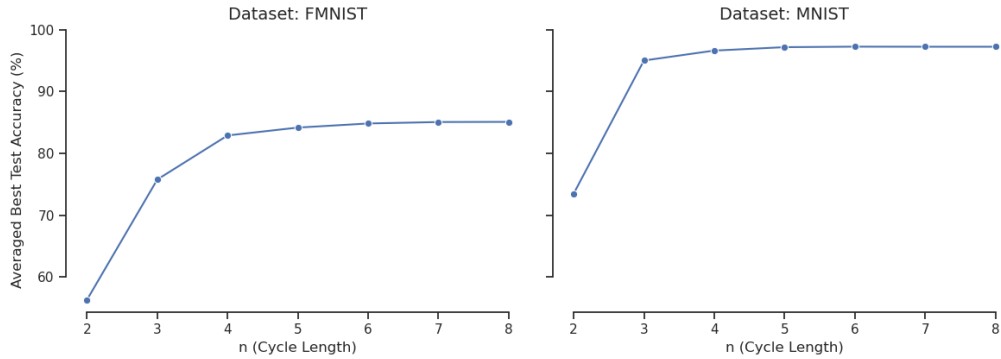

(a) Effect of cycle length n (see Eq. 8) on test accuracy with fixed decay (a=1.0). Performance increases with n on both MNIST and Fashion-MNIST, though returns diminish for sufficiently large n.

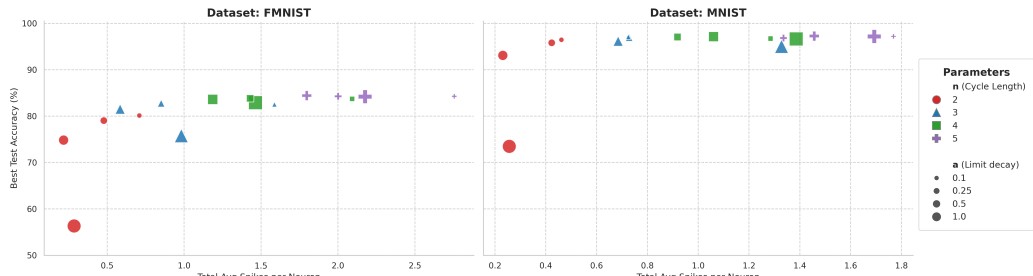

(b) Trade-off between test accuracy and communication cost (spikes). Varying the decay parameter a (marker size) can improve accuracy, particularly for smaller cycle lengths n (marker shape/color), at the cost of increased spike activity.

Figure 2: Analysis of scheduler hyperparameters n and a on MNIST and Fashion-MNIST. The top figure isolates the effect of n, while the bottom figure shows the interplay between n, a, accuracy, and spike cost.

cycle length n already provides sufficient timesteps for the network's states to converge with high precision. In this regime, the fine-tuning offered by the adaptive decay (a¡1.0) becomes redundant, as the inherent precision of the long spike train is already maximal for the task.

## 7.5 HYPERPARAMETER SELECTION HEURISTICS

Choosing optimal hyperparameters for DiffPC, as with many complex models, is a non-trivial task without a closed-form solution. However, we have identified several heuristics that provide a strong starting point for tuning the network for a new task.

From standard predictive coding theory, a functional network requires a minimum number of relaxation steps, typically at least twice the depth of the network, to allow information to propagate fully between the input and output layers (see e.g. A.3.2 of Innocenti et al. (2025b)). This principle provides a useful guideline for the minimum number of timesteps required. For DiffPC, we offer the following more specific guidance.

**Choosing m** For the cyclic scheduler, a robust choice is m=2 when using a clipped activation function like ReLU6. The parameter m sets the initial and largest threshold value in a cycle, which is $2^m$. If this value is significantly larger than the maximum possible activation (e.g., 6 for ReLU6), the initial timesteps will generate no spikes, as the difference $|x_T - x_A|$ will never exceed the threshold. Setting m=2 yields an initial threshold of 4, which is on the same order of magnitude as the activation range, ensuring that the spike generation process is active from the beginning of the cycle.

**Choosing n and a**    The parameters n and a jointly control the trade-off between runtime, communication cost, and precision. A practical approach to tuning them is a two-step process:

1. **Find an effective cycle length n.** First, set a=1.0 (disabling adaptive decay) and incrementally increase n while monitoring test accuracy. Continue until performance saturates, establishing a baseline for the required precision.

2. **Optimize for efficiency.** Once a saturation point $n_{\text{sat}}$ is found, one can attempt to reduce the cycle length to $n_{\text{new}} = n_{\text{sat}} - k$ for some small integer k, thereby reducing runtime. To compensate for the potential loss of precision, the decay factor can be set to $a \approx 1/2^k$.

The reasoning for this two-step process is as follows. Reducing the cycle length by k steps removes the k timesteps that have the smallest, and therefore most precise, threshold values. To compensate, a smaller value of a is used to scale down the entire threshold schedule within the new, shorter cycle. The heuristic $a \approx 1/2^k$ is specifically chosen because it ensures that the final, smallest threshold in the new $n_{\text{new}}$-step cycle is approximately equal to what the final threshold was in the original $n_{\text{sat}}$-step cycle with a=1.0. This approach aims to recover the necessary representational fidelity while benefiting from a shorter runtime.

## 8   TABLE OF NOTATION

Table 6 summarizes the mathematical symbols used in the Difference Predictive Coding (DiffPC) algorithm.

Table 6: Nomenclature and Symbols

| Symbol | Description |
|---|---|
| *Shared Variables (Standard PC & DiffPC)* | |
| $l$ | Layer index, $l \in \{0, \dots, L\}$. |
| $\mathbf{W}^l$ | Synaptic weight matrix connecting layer $l - 1$ to $l$. |
| $\mathbf{x}_F$ | **Forward Prediction**. The prediction generated by the previous layer. |
| $\mathbf{x}_T$ | **Target Activity**. The ideal state calculated to minimize prediction energy. |
| $\epsilon$ | **Prediction Error**. The difference between target and prediction $(\mathbf{x}_T - \mathbf{x}_F)$. |
| $\gamma(t)$ | Inference learning rate at time $t$. |
| *DiffPC-Specific States (Spiking Implementation)* | |
| $\mathbf{x}_A$ | **Actual Activity**. The discrete state that tracks the shared target $\mathbf{x}_T$, updated via spikes. |
| $\mathbf{s}_A$ | **Activity Spikes**. Ternary spikes $\{-1, 0, 1\}$ communicating changes in $\mathbf{x}_A$. |
| $\mathbf{e}_T$ | **Target Error**. The local error variable (functionally equivalent to $\epsilon$ in this context). |
| $\mathbf{e}_A$ | **Actual Error**. The discrete state that tracks $\mathbf{e}_T$, updated via spikes. |
| $\mathbf{s}_e$ | **Error Spikes**. Ternary spikes communicating changes in the error state. |
| $\mathbf{e}_B$ | **Backward Error**. The error signal accumulated from layer $l + 1$. |
| *Scheduler & Thresholds* | |
| $T_\theta(t)$ | Adaptive firing threshold at time $t$. |
| $m$ | Scheduler magnitude parameter (sets max threshold $2^m$). |
| $n$ | Scheduler cycle length (periodicity of the steps). |
| $a$ | Decay factor for the cyclic decay scheduler. |

# 9 EVENT-DRIVEN RESPONSE TO INPUT CHANGES

To demonstrate the event-driven nature of our method, we conducted an experiment to measure the network's spiking activity in response to a changing input. We presented a static input image from the test set to a trained DiffPC network and monitored the total number of activity spikes ($s_A$) across all layers over time. After an initial period of 25 timesteps, we introduced an abrupt change by shifting the input image by a single pixel.

The results, averaged over 1000 different input images, are shown in Figure 3. Initially, there is a burst of spiking activity as the network processes the new image. This activity quickly subsides, and the network becomes nearly silent as its internal state converges to a stable representation of the static input. At timestep 25 when the input is shifted, the network immediately responds with another burst of spikes, which then decays as it settles into a new stable state.

This behavior highlights a key feature of DiffPC: computation is performed only when necessary to process new or changed information. For applications where inputs may remain static for periods of time, this event-driven property suggests the potential for energy savings by eliminating redundant processing, making the approach highly suitable for energy-constrained neuromorphic hardware.

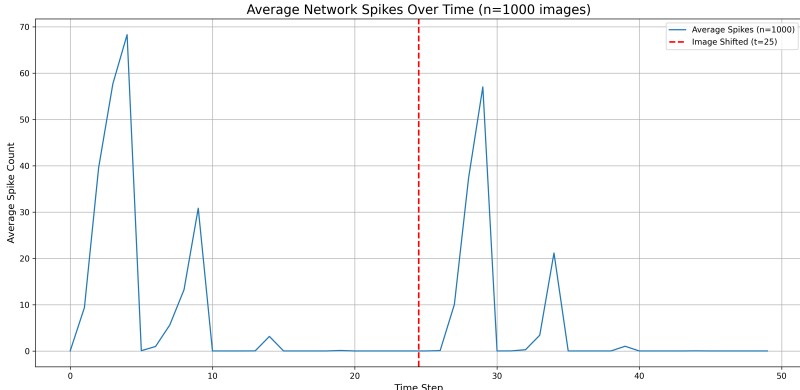

Figure 3: Event-driven spiking in a trained DiffPC network. The network shows an initial burst of spikes when an image is presented, then falls silent. A second burst of activity is triggered precisely at timestep 25, when the input image is shifted by one pixel, demonstrating that the network only computes in response to change.

## REPRODUCIBILITY STATEMENT

We are committed to ensuring the reproducibility of our research. The complete PyTorch implementation of the Difference Predictive Coding algorithm, including model architectures, schedulers, and training procedures, will be made available as supplementary material. Our core method is detailed in Section 7.2, with a step-by-step breakdown provided in the Appendix. The source code includes the exact configurations and hyperparameters used to generate all reported results, including classification accuracy on MNIST (Table 1) and Fashion-MNIST (Table 2), and the communication efficiency analysis (Table 3). Further, the codes used to generate the CIFAR-10 results will also be made public. All experiments were conducted using standard public datasets, and the specific data processing pipelines are explicitly defined within our implementation.

## THE USE OF LARGE LANGUAGE MODELS (LLMS)

In the preparation of this manuscript, Large Language Models (LLMs) were utilized as a general-purpose assistive tool. The authors take full responsibility for all content, ensuring its scientific accuracy and originality. The specific roles of the LLMs are outlined below:

- **Writing Assistance:** LLMs were employed to improve the language and clarity of the manuscript. This included refining sentence structures, correcting grammatical errors, and ensuring overall readability. The core scientific ideas, arguments, and conclusions presented are entirely the work of the authors.

- **Literature Discovery:** LLMs were used as a tool to aid in the literature review process by suggesting potentially related academic papers and summarizing established concepts. All works cited in this paper were subsequently retrieved, read, and critically evaluated by the authors to verify their relevance and accuracy.

- **Coding Support:** LLMs assisted in the software development process by generating boilerplate code, helping to debug specific code segments, and suggesting algorithmic optimizations. The overall design of the experiments, the core logic of the implementation, and the final analysis were conceived and performed by the authors.

