# OpenReview forum: "Difference Predictive Coding for Training Spiking Neural Networks"
_ICLR.cc/2026/Conference — ICLR 2026 Poster_

### Official Review · Reviewer_htVv · 2025-10-23

**Soundness:** 3
**Presentation:** 2
**Contribution:** 2
**Rating:** 4
**Confidence:** 3

**Summary:**

This paper proposes DiffPC, a spike-native local learning algorithm for SNNs. DiffPC fits the Predictive Coding algorithm into SNNs, replacing floating-point messages with sparse ternary spikes, providing spike-compatible target and error updates, and employing adaptive threshold schedules for event-driven operation. Experimental results have shown the effectiveness and energy efficiency of DiffPC.

**Strengths:**

1. The proposed algorithm only needs local information propagation, which is much more compatible with SNNs than BP-based algorithms.
2. The proposed algorithm propagates information through spikes during training and inference, which suits the characteristic of SNNs.

**Weaknesses:**

1. There are many symbols in this paper, and some are not clearly explained, making it difficult to follow.
2. The writing of this paper is not clear enough. See Questions for details.

**Questions:**

1. Could you please introduce Predictive Coding more clearly? How to initialize the parameters such as $x_T$? From Algorithm 1, it seems there are $x_T, x_F, x_A$ for the neuron state. However, only $x_T$ and $x_F$ is introduced in Section 3.2. Does $x_A$ appear in PC? Also in Algorithm 1, the step of updating weights seems not included.
2. I recommend the authors to place a table illustrating the meaning of each symbol. If the room is not enough in main body, you can put it in appendix.
3. What is the intuitive behind the PC method? How the network is trained to predict classification result through PC? Please give more detailed intuition.
4. What is the number of 'the number of approximation steps *n*' in line 426? Is it the number of optimization steps?
5. There are two 'Energy efficiency' in 5.2 RESULTS. Please change one of them. In line 355, a full stop is missing.
6. In Figure 2, why there are two peaks of spike activity after input change? Could the authors give some intuition?

---

> ### Author Response · Authors · 2025-11-21
>
> We thank the reviewer for recognizing the strengths of DiffPC as a spike-native, local learning algorithm that is highly compatible with neuromorphic characteristics. We appreciate the feedback regarding the clarity of the presentation and symbols and recognize that the complexity of the algorithm requires great precision in presentation. We have revised the manuscript to improve readability and address the specific questions raised.
>
> ### Responses to Specific Points
>
> > **Could you please introduce Predictive Coding more clearly? ... From Algorithm 1, it seems there are [states]... However, only [standard symbols] are introduced in Section 3.2. Also in Algorithm 1, the step of updating weights seems not included.**
>
> **DiffPC clarity:** We have have added extra detail into the description of the DiffPC algorithm which is located in the Appendix section 6.1.
>
> **Initialization:** In our implementation, internal state parameters (like $\vec{x}_A$) are initialized to zero. The target activity variables $\vec{x}_T$ at the input and output layers are clamped to the input data and target labels, respectively. Crucially, the output layer is clamped only *after* the initial forward propagation phase, ensuring the network establishes a prediction baseline before error minimization begins. We have added a subsection to the paper explicitly detailing this initialization sequence.
>
> **Weight Updates:** Thank you for notifying us that Algorithm 1 does not include the weight update step. The weights are updated using the locally accumulated values according to the rule: $\Delta W \propto - e_T \cdot \phi(x_T)^T$. We have added this explicit update step to the manuscript.
>
> > **I recommend the authors to place a table illustrating the meaning of each symbol.**
>
> This is an excellent suggestion to improve readability. We have added a comprehensive **Table of Notation** to the Appendix (Section 8). It categorizes symbols into "Shared Variables (Standard PC & DiffPC)," "DiffPC-Specific States (Spiking Implementation)," and "Scheduler & Thresholds," providing clear definitions for every variable used in the algorithms.
>
> > **What is the intuitive behind the PC method? How the network is trained to predict classification result through PC?**
>
> To ensure this intuition is communicated clearly without overcrowding the main methodology section, we have added a dedicated **INTUITION OF PREDICTIVE CODING TRAINING** section to the Appendix (Section 7.1). This section breaks down the process into: 1) **Prediction** (Forward pass), 2) **Constraint** (Clamping outputs), 3) **Relaxation** (Errors flowing back to shift hidden states), and 4) **Learning** (Updating weights to facilitate those states).
>
> > **What is the number of 'the number of approximation steps n' in line 426? Is it the number of optimization steps?**
>
> We have clarified this definition in Section 4.2 where it is first introduced. $n$ refers to the **cycle length** of the threshold scheduler (the quantization period). It determines how many timesteps are used to refine the spike-based approximation of the continuous signal during the settling process. It is *not* the number of optimization (weight update) steps; rather, it defines the duration of the integration window within a single training iteration.
>
> > **There are two 'Energy efficiency' in 5.2 RESULTS. Please change one of them. In line 355, a full stop is missing.**
>
> Thank you for catching these editorial errors. We have corrected the duplicate section header and fixed the missing punctuation.
>
> > **In Figure 2, why there are two peaks of spike activity after input change? Could the authors give some intuition?**
>
> Figure 2 illustrates the sequential propagation of activity in an event-driven system.
>
> 1.  **Initial Cascade ($t=0-15$):** When the image is presented, the input layer bursts first. This activity propagates to the hidden layers, causing a secondary wave of spikes as they adjust their internal states ($\vec{x}_A$) to match the incoming predictions. As the network settles into a consistent state, the difference $|\vec{x}_T - \vec{x}_A|$ drops below the threshold, and the network falls silent.
> 2.  **Input Shift ($t=25$):** We shift the input image by 1 pixel. This creates a sudden, large prediction error at the input. The network immediately bursts again to correct for this discrepancy, propagating the new information through the layers.
>
> This demonstrates that DiffPC only expends energy (computes) when there is new information or a change in the environment to process.

---

> > ### Comment · Reviewer_htVv · 2025-11-25
> >
> > Thank you for making the paper clearer. However, from the INTUITION OF PREDICTIVE CODING TRAINING, I don't see the key difference between DiffPC and backpropagation. DiffPC also has a Backward Error Flow, which not only relies on local information. Could the authors explain more about the difference between DiffPC and backpropagation?

---

> > > ### Author Response · Authors · 2025-11-25
> > >
> > > We thank the reviewer for the follow-up question. While both algorithms involve a backward flow of information, they differ fundamentally in the nature of that flow and the locality of the weight updates.
> > >
> > > **1. Relaxation via Bidirectional Flow vs. Gradient Calculation**
> > >
> > > *   **Backpropagation:** The backward pass is unidirectional and it calculates the gradients of the weights with respect to a global loss function. The neuron activities (states) remain fixed after the forward pass.
> > > *   **DiffPC:** The learning process relies on a **Relaxation** phase characterized by **bidirectional** information flow. Predictions flow forward (top-down) while errors flow backward (bottom-up). The network iteratively updates its internal activities to reconcile these opposing signals. The network dynamically changes its activities to minimize the error *before* any weight update occurs. Then, the weights are changed in such a way that these activities would be produced without errors in the future.
> > >
> > > **2. Local Weight Updates**
> > >
> > > *   **Backpropagation:** To update a weight, the algorithm needs to know how that specific weight affects the global loss, requiring the transport of gradients through the entire chain of layers.
> > > *   **DiffPC:** Once the relaxation process has adjusted the neuron activities, the weight update is **strictly local**. As shown in Algorithm 1, the weight change depends *only* on the immediate pre-synaptic activity and the immediate post-synaptic error.
> > >
> > > To summarize these distinctions, we present the following comparison table:
> > >
> > > | Aspect | Backpropagation | Difference Predictive Coding (DiffPC) |
> > > | :--- | :--- | :--- |
> > > | **Information Flow** | **Unidirectional:** Backward flow of gradients from output to input. | **Bidirectional:** Predictions flow forward, errors flow backward. |
> > > | **Neuron States** | Fixed during the backward pass. | **Dynamic:** Activities change iteratively to minimize local error (Relaxation). |
> > > | **Update Mechanism** | **Global:** Depends on the global loss function (Chain Rule). | **Local:** Depends only on locally available pre- and post-synaptic states. |
> > > | **Efficiency Context** | Efficient in standard floating-point (FP) implementations. | While computationally more expensive than BP in FP simulations, the local nature of PC allows for **highly efficient, event-driven implementation in SNNs**, drastically reducing communication requirements. |
> > >
> > > We will revise the **INTUITION OF PREDICTIVE CODING TRAINING** section in the Appendix to clarify the bidirectionality of PC.

---

> > > > ### Comment · Reviewer_htVv · 2025-11-26
> > > >
> > > > Your arguments still cannot convince me.
> > > >
> > > > 1. For information flow, BP also has a forward pass and a backward pass, which can be viewed as bidirectional.
> > > > 2. Update mechanism: assume there are $L$ layers in the SNN, in BP, the gradients propagate direction is loss function -> activity in layer $L$ -> activity in layer $L-1$ -> ... -> activity in layer $1$, and activity in layer $L$ -> weight between layer $L-1$ and $L$, ..., activity in layer $1$ -> weight between input and layer $1$. Under the assumption that gradients have already been passed to activities, the propagation from activity to weight is local.

---

> > > > > ### Author Response · Authors · 2025-11-26
> > > > >
> > > > > You make a fair point, and we want to align on this interpretation first. We agree that if the error signal has already arrived at the layer, the final weight update step in Backpropagation is indeed local. We also agree that both algorithms fundamentally rely on moving information in two directions (forward and backward). However, the reason these two algorithms are distinct isn't about the direction of the flow, but rather the behavior of the neurons while that flow is happening.
> > > > >
> > > > > This distinction is central to DiffPC. Because the states of the neurons are "moving" (changing states) to minimize error, we can treat this as an event-driven process which occurs in each layer independent from others. Then we let the neuron states converge (settle), transmitting spikes only when they change state significantly. The weights are then updated using these settled states, rather than by computing the gradient of a global loss.
> > > > >
> > > > > In the theoretical literature (e.g., Millidge et al., ICLR 2023 [ https://openreview.net/forum?id=ZCTvSF_uVM4 ]), this regime is called **Prospective Configuration**. It is mathematically demonstrated that in this regime, PC does not merely approximate Backpropagation gradients, but performs a distinct constrained **Expectation-Maximization** process. Consequently, DiffPC and Backpropagation follow different trajectories through the weight space, even while minimizing the same global loss.

---

> > > > > > ### Comment · Reviewer_htVv · 2025-11-28
> > > > > >
> > > > > > OK. There seems to be some theoretical difference between BP and PC. However, I think the authors should emphasize the advantages of PC to BP (I recommend authors find some quantized indicators), which is unclear now. The disadvantage of PC is obvious, for instance, in this work, PC can only train shallow SNN, while BP can train deep ones.

---

> > > > > > > ### Author Response · Authors · 2025-11-28
> > > > > > >
> > > > > > > We thank the reviewer for encouraging a clearer exposition of predictive coding's empirical benefits beyond backpropagation. In recent empirical work, PC models have demonstrated several practical advantages.
> > > > > > >
> > > > > > > First, PC enables **continual online learning**: under streaming (batch-size one) training, PC retains performance on earlier tasks significantly better than BP, showing reduced interference and slower forgetting in sequential task settings [1](Fig. 4D). Second, PC exhibits strong **multitask learning capability**, efficiently adapting to task switches and label remappings with minimal loss of prior knowledge, outperforming BP under periodic concept drift scenarios [1](Fig. 4F).
> > > > > > >
> > > > > > > Third, PC demonstrates greater **sample efficiency**. Recent work has shown that PC approximates implicit stochastic gradient descent, effectively leveraging second-order (curvature) information during learning [2]. This allows PC to achieve faster loss reduction and convergence compared to standard backpropagation, particularly in the small-batch regimes typical of biological or online learning setups [2](Fig. 3).
> > > > > > >
> > > > > > > Fourth, PC networks are more robust to out-of-distribution (OOD) inputs: as energy-based models, they assign significantly higher free-energy values to unfamiliar inputs, achieving strong OOD detection "out of the box" without retraining [3](Fig. 25).
> > > > > > >
> > > > > > >
> > > > > > > Finally, regarding **scalability to deep networks**: It is true that historically standard PC has faced challenges with depth, already starting to lose performance in as few as seven layers [3](Table 1). However, recent emerging work such as $\mu$-PC [4] demonstrates that PC can scale to 100+ layer ResNets when inference dynamics are stabilized. Since DiffPC is designed as a faithful spike-based discretization of PC, changing only the message representation, not the underlying objective, we hypothesize that these stabilization techniques will transfer directly. Naturally, verifying this requires empirical validation. A complete scaling study would examine the fidelity of the DiffPC–PC approximation as depth increases. Key analyses include: (i) the average layer-wise deviation between DiffPC and continuous PC states; (ii) the evolution of this deviation as a joint function of depth and inference steps; and (iii) the emergence of iso-error curves—i.e., whether deeper networks require longer spike-based communication windows to maintain a fixed approximation error. Additionally, identifying which emerging stabilization technique optimally complements the discrete DiffPC constitutes a dedicated study, which we highlight as a direction for future research.
> > > > > > >
> > > > > > >
> > > > > > > **References:**
> > > > > > >
> > > > > > > [1] Inferring neural activity before plasticity, Nature Neuroscience (2023). https://www.nature.com/articles/s41593-023-01514-1
> > > > > > >
> > > > > > > [2] Understanding and Improving Optimization in Predictive Coding Networks, AAAI (2024). https://ojs.aaai.org/index.php/AAAI/article/view/28954
> > > > > > >
> > > > > > > [3] Benchmarking Predictive Coding Networks – Made Simple, arXiv (2024). https://arxiv.org/abs/2407.01163
> > > > > > >
> > > > > > > [4] $\mu$-PC: Scaling Predictive Coding to 100+ Layer Networks, arXiv (2025). https://arxiv.org/abs/2505.13124

---

### Official Review · Reviewer_8Q9N · 2025-10-24

**Soundness:** 2
**Presentation:** 2
**Contribution:** 2
**Rating:** 2
**Confidence:** 4

**Summary:**

The primary contribution of DiffPC lies in being the first predictive coding framework trained fully through spike-based event-driven mechanisms, effectively bridging the gap from theoretical feasibility to hardware-ready neuromorphic deployment. Nevertheless, the current experimental scope remains limited, and the theoretical analysis could be further strengthened. More extensive evaluations on diverse tasks and real neuromorphic hardware measurements are needed to substantiate the method’s generality and energy-efficiency advantages.

**Strengths:**

1. This work is the first to fully discretize both target updates and error transmission in predictive coding into sparse ternary spike events, eliminating the need for floating-point communication and external ANN-based training pipelines.

2. On the MNIST benchmark, DiffPC achieves a test accuracy of 99.3%, comparable to standard predictive coding and backpropagation, while delivering more than a 5,000-fold improvement in communication efficiency.

**Weaknesses:**

1. The Introduction provides an insufficient and somewhat disorganized treatment of the importance of local learning. I recommend a thorough rewrite of this section to clarify the narrative and strengthen the conceptual motivation.

2. The Introduction lacks a high-level articulation of the motivation for proposing DiffPC. Based solely on this section, readers cannot clearly infer what the authors aim to do, what has been accomplished, or the scholarly significance of the contributions.

3. The experimental evaluation is severely limited, and results on MNIST and Fashion-MNIST are inadequate to substantiate the method’s effectiveness. At minimum, an evaluation on CIFAR-10 is recommended.

4. In the Related Works section, citations to similar training frameworks should be mapped one-to-one with the corresponding methods and claims.

5. There are multiple presentation issues: a symbol error at the end of line 195; missing punctuation after Equation (11); inconsistent decimal precision in the “Acc.” column of Table 2; and numerous other minor formatting and symbol inconsistencies, likely due to a rushed preparation. Please correct these errors throughout.

**Questions:**

1. Lines 156–164 state that four innovations are introduced. However, the first contribution—ternary spiking—has already been widely adopted in SNN research. The third contribution—dynamic thresholding—has also been extensively explored in prior SNN literature. Regarding the fourth contribution, the near-exact PCN approximation resembles a conversion or distillation process; therefore, clarification is needed on whether it should be viewed as a form of teacher–student distillation rather than an independent methodological innovation.

2. The manuscript argues that surrogate-gradient-based backpropagation is biologically implausible due to reliance on global gradients, whereas DiffPC uses local plasticity and is thus more brain-like. However, validating DiffPC only on MLP architectures does not substantiate this claim. The experiments do not reveal whether a fully local learning mechanism can scale to larger networks. More importantly, when every layer is trained exclusively through DiffPC, the paper does not analyze how network depth or size influences final model performance.

3. Although the paper emphasizes its relevance to neuromorphic tasks, no temporal benchmarks are included. Evaluation on event-driven datasets would be necessary to support this claim.

4. The authors are encouraged to provide real power measurements on Loihi 2 hardware, and further discuss the quantitative advantages over PC-SNN and surrogate-gradient-based SNN training, along with a clearer articulation of the fundamental differences.

---

> ### Author Response · Authors · 2025-11-21
>
> We sincerely thank the reviewer for their analysis of our manuscript. We value the feedback regarding the organization of the introduction, the need for wider experimental validation, and the theoretical questions regarding scalability. We have addressed these weaknesses directly in the revised manuscript.
>
> ### Responses to Specific Points
>
> > **The Introduction provides an insufficient and somewhat disorganized treatment of the importance of local learning... readers cannot clearly infer what the authors aim to do.**
>
> We agree with the reviewer that the original introduction was not sufficiently clear in articulating the specific motivation for our work and the gap we aim to fill. As such we have rewritten the introduction to clarify the narrative.
>
> The revised Introduction now follows a structured logic to better guide the reader:
> *   **The Problem with Backprop:** It begins by establishing the biological and hardware constraints that make Backpropagation suboptimal (global error signals) for some hardware.
> *   **The Promise of PC:** It introduces Predictive Coding (PC) as a solution via **local learning**, making it suited for distributed neuromorphic substrates.
> *   **The Crucial Gap:** It explicitly identifies the bottleneck of standard PC: despite being local, it relies on *dense, iterative floating-point communication*, which incurs a high computational overhead that negates the energy benefits of event-driven hardware.
> *   **The Solution (DiffPC):** It positions DiffPC as the necessary bridge that combines the local learning rules of PC with the *communication sparsity* of SNNs. By replacing dense messages with sparse ternary spikes, DiffPC realizes the efficiency potential that PC theoretically offers.
>
> We believe this restructured narrative clearly defines the problem scope, the logic of our approach, and the significance of our contribution.
>
> > **The experimental evaluation is severely limited... At minimum, an evaluation on CIFAR-10 is recommended.**
>
> We appreciate the feedback regarding the scope of our initial evaluation. To demonstrate the method's applicability to more complex tasks, we have followed the recommendation to expand our experimental section by implementing a convolutional variant of DiffPC (architecture: 10C5S2-5C5S2-50FC-30FC-10FC) and evaluating it on the **CIFAR-10** dataset.
>
> *   **Results:** The convolutional DiffPC network achieves a test accuracy of **65.6%**, which outperforms a standard backpropagation baseline (63.5%) using the exact same architecture (Table 4 in revised paper).
> *   **Efficiency:** Crucially, this performance is achieved while maintaining high communication sparsity (approx. 1.9 bits/neuron).
>
> These new results confirm that the DiffPC learning rule scales effectively to convolutional architectures.
>
> > **In the Related Works section, citations to similar training frameworks should be mapped one-to-one with the corresponding methods and claims.**
>
> We thank the reviewer for this suggestion to improve clarity. We have revised the Related Works section to improve descriptions of cited frameworks and their specific methodologies.

---

> ### Author Response · Authors · 2025-11-21
>
> > **Validating DiffPC only on MLP architectures does not substantiate [the biological plausibility] claim... the paper does not analyze how network depth or size influences final model performance.**
>
> We thank the reviewer for raising the question of scalability. DiffPC is intentionally designed as a spike-based communication protocol for predictive coding (PC). After communication, the computation performed is identical to standard PC. In our experiments we verified empirically that DiffPC states closely track those of standard PC, confirming that DiffPC constitutes a faithful discretization of the underlying PC dynamics. Because of this equivalence, it is natural to investigate whether DiffPC inherits the scalability properties of PC itself. Scaling PC is an active research topic. Recent work (e.g. *$\mu$PC: Scaling Predictive Coding to 100+ Layer Networks (2025)*) shows that the main obstacles to depth scaling are ill-conditioned inference dynamics and vanishing/exploding feedforward signals. Importantly, that work demonstrates that appropriate parameterization enables training 100+ layer ResNets with PC, showing that deep predictive-coding networks can indeed scale when these issues are addressed. Since DiffPC does not modify the PC objective or inference mechanism—only the representation and transmission of messages—it is reasonable to expect that the same stabilization techniques transfer directly to DiffPC. Naturally, this needs to be empirically validated. A complete scaling study would examine how the DiffPC–PC approximation behaves as depth increases. Natural metrics include: (i) the average layer-wise deviation between DiffPC and continuous PC states as a function of depth; (ii) how this deviation changes as a joint function of depth and number of inference steps; and (iii) whether iso-error curves emerge—i.e., whether deeper networks require longer spike-based communication windows to maintain a fixed approximation error. These analyses constitute a dedicated study of discretized PC inference dynamics. Since DiffPC preserves the PC optimization and only alters the communication format, we expect its behavior to mirror that of PC up to discretization effects. A systematic evaluation of these depth-dependent scaling behaviors is beyond the scope of the present submission, but we will highlight it explicitly as future work in the conclusions section.
>
> > **Clarification is needed on whether [the approximation] should be viewed as a form of teacher–student distillation rather than an independent methodological innovation.**
>
> While the convergence dynamics share similarities with distillation, DiffPC operates fundamentally differently from a standard teacher-student setup.
> In standard distillation, a pre-trained, static "teacher" network provides soft targets to a student. In DiffPC:
> 1.  There is no external teacher. The "target" ($\vec{x}_T$) is a dynamic variable internal to the network itself, which evolves online based on the error inputs.
> 2.  The update rules are derived directly from minimizing the variational free energy of the system using discrete steps, rather than mimicking a separate network's output.
>
> Regarding the components (ternary spikes, thresholds): We agree these elements exist individually in SNN literature. Our contribution is the specific mathematical derivation that unifies them into a coherent *learning rule* that relieves the communication bottleneck of Predictive Coding. We show how to map the gradients of the PC energy function into these discrete events.

---

> ### Author Response · Authors · 2025-11-21
>
> > **The authors are encouraged to provide real power measurements on Loihi 2 hardware... No temporal benchmarks are included.**
>
> **Regarding Power:** As we are validating our approach via a Loihi 2 simulation (LAVA) rather than physical measurements on a generic setup, we cannot accurately report calibrated energy numbers (Joules/inference) without physical access to the specific Loihi 2 board, which is restricted. Instead, we frame our efficiency claims strictly in terms of **Communication Sparsity** (Bits per Neuron). Since data movement is the dominant energy cost in digital accelerators, this is a important hardware-agnostic measure for energy efficiency.
>
> **Regarding Temporal Benchmarks:** We agree that event-driven datasets are a natural fit for neuromorphic validation. However, our primary focus in this work was establishing that DiffPC can train spiking networks on standard benchmarks using purely local rules. Regarding the method's temporal capabilities, we highlight the experiment in Section 9 (Figure 3), which explicitly tests the network's response to dynamic input changes. By shifting the input image during simulation, we demonstrate that the network remains silent during steady states and only generates spikes when information changes. This confirms that DiffPC intrinsically exploits temporal sparsity, operating in a fully change-driven manner.
>
> > **There are multiple presentation issues... Please correct these errors throughout.**
>
> We apologize for these oversight errors and have gone through the manuscript to find such mistakes.

---

> > ### Comment · Reviewer_8Q9N · 2025-11-24
> > **I am willing to raise my score to 4.**
> >
> > Thank you for your response. I acknowledge the authors' efforts in preparing the rebuttal, which has addressed some of my concerns. However, I still believe that this work would be more convincing if extended to deeper SNNs. I feel that my question "Validating DiffPC only on MLP architectures does not substantiate [the biological plausibility] claim... the paper does not analyze how network depth or size influences final model performance." cannot be adequately answered solely through theoretical explanations, as this represents an important contribution of this paper. Additionally, I still do not consider the ternary spiking neurons and dynamic thresholds to be valid contributions of this work.
> >
> > Nevertheless, the authors have supplemented their work with experiments on CIFAR-10 and have substantially revised their manuscript based on my suggestions. I believe the current version is more readable and more persuasive. I am willing to raise my score to 4. I wish the authors all the best.

---

> > > ### Author Response · Authors · 2025-11-24
> > >
> > > We thank you for raising the score to 4 and for your constructive feedback, which has helped us significantly improve the manuscript’s clarity and focus.
> > >
> > > We understand your remaining reservations regarding validation on deeper architectures and the novelty of specific components. We would like to offer the following thoughts to clarify our position:
> > >
> > > **1. On Scalability and Deep Networks**
> > >
> > > We agree with the premise that for a learning rule to be fully established, it must eventually scale to deep architectures. However, we believe that the challenge of scaling to networks like ResNet-18 currently lies with the underlying Predictive Coding framework itself, rather than our spiking formulation.
> > >
> > > As shown in benchmarking literature (e.g., Pinchetti et al. (https://arxiv.org/abs/2407.01163)), standard floating-point PC struggles with deeper networks, often seeing accuracy drop drastically on deeper VGG and ResNet architectures without auxiliary stabilization techniques (such as those introduced in the very recent $\mu$-PC). Since DiffPC is designed to be a faithful spike-based discretization of standard PC, it naturally inherits these intrinsic algorithmic characteristics.
> > >
> > > We therefore view the stabilization of PC convergence at depth as a research direction orthogonal to our contribution. Our specific goal in this work was to solve the *communication* problem, demonstrating that if a PC network can be trained, it can be done with sparse spikes, reducing data movement by orders of magnitude. We believe that applying DiffPC to deeper networks is the logical next step, but one that relies on first finding which of these emerging stabilization techniques are the best fit with our DiffPC algorithm and how exactly they should be optimally implemented.
> > >
> > > **2. On Novelty**
> > >
> > > We also appreciate your distinction regarding the network components. We fully acknowledge that ternary spikes and dynamic thresholds are well-established mechanisms in the SNN literature, and will ensure the final manuscript explicitly credits the origins of these components to avoid any overclaiming, focusing instead on their novel application within this specific learning framework designed to alleviate the communication bottleneck of standard PC.
> > >
> > > We are very interested in ensuring this work meets the bar for acceptance. Is there a specific discussion or analysis we could add to the final version that might help alleviate your remaining concerns?
> > >
> > > Thank you again for your time and guidance.

---

### Official Review · Reviewer_VdSN · 2025-10-26

**Soundness:** 3
**Presentation:** 3
**Contribution:** 2
**Rating:** 4
**Confidence:** 4

**Summary:**

The paper presents a spike-based version of the Predictive Coding (PC) for training, equivalent to standard PC and called DiffPC. The method consists of tracking the changes of the continuous signal by sending "corrective" ternary spikes, by iteratively adjusting the "weight" of the spikes using an adaptative threshold. Equivalence with PC is empirically demonstrated on MNIST and Fashion-MNIST benchmarks.

**Strengths:**

- The paper is well written and easy to read;
- The DiffPC implements a relatively classic spike coding method (based on tracking a continuous signal with spike corrections) that has already been successfully applied to backpropagation as well;
- The method is mathematically equivalent to standard PC, modulus the final threshold value approximation.

**Weaknesses:**

- The cyclic scheduler equations are not very well explained. Parameter m is not defined (equation 8).
- The DiffPC relies on scheduling hyperparameters that may be hard to tune to get the best efficiency vs accuracy compromise. A discussion about this is missing from the main paper (although mentionned in the supplementary material);
- Although some data on energy efficiency are provided, it is not clear of they translates to real world implementation efficiency.

**Questions:**

The paper lacks a conclusion/perspective section, is it on purpose?
Regarding implementation efficiency, could the author provide some insight or benchmark that compares standard PC and DiffPC?

Overall, the quality of the paper could still be improved: more explanation/insight on the method scheduling could be provided. The paper also lacks a conclusion. Table 4 and related content could be improved, perhaps with a graph better illustrating the convergence between PC and DiffPC as the threshold becomes smaller and an absolute scale or a relative one. But the mean absolute difference is not very meaningful...

---

> ### Author Response · Authors · 2025-11-21
>
> We thank the reviewer for their positive assessment of the manuscript's clarity and soundness. We appreciate the constructive feedback regarding the scheduler definitions and the presentation of results. We have addressed these weaknesses directly in the revised manuscript.
>
> ### Responses to Specific Points
>
> > **The cyclic scheduler equations are not very well explained. Parameter m is not defined (equation 8).**
>
> We apologize for this oversight. We have revised Section 4.2 to provide a clarified definition of the scheduler parameters.
> Specifically, $m$ defines the **initial magnitude** of the threshold schedule. It serves to align the dynamic range of the quantization with the activation range of the network. For example, when using clipped activations (like ReLU6), setting $m$ appropriately ensures the initial threshold covers the full range of possible errors, preventing saturation at the start of a cycle. We also further motivated the cyclic scheduler by adding Theorem 4.1.
>
> > **The DiffPC relies on scheduling hyperparameters that may be hard to tune... A discussion about this is missing from the main paper.**
>
> We agree that practical guidance for hyperparameter selection is essential for reproducibility. To address this, we have added a **HYPERPARAMETER ANALYSIS** and **HYPERPARAMETER SELECTION HEURISTICS** sections to the Appendix.
> *   We performed an ablation study on the cycle length ($n$) and decay factor ($a$). The results show a clear trade-off: increasing $n$ improves precision (and accuracy) but increases latency.
> *   We provide a "Rule of Thumb": Maximize $n$ until accuracy saturates (typically $n \approx 4-8$), then reduce $n$ to improve efficiency while tuning the decay factor $a$ to maintain representational fidelity.
>
> This addition provides the missing insight into the efficiency-vs-accuracy compromise requested by the reviewer.
>
> > **Regarding implementation efficiency, could the author provide some insight or benchmark that compares standard PC and DiffPC?**
>
> To provide a concrete comparison, we have expanded our efficiency analysis using the CIFAR-10 benchmark.
> *   **Standard PC:** Relies on dense, floating-point communication. It must transmit both predictions ($x_F$) and errors ($\epsilon$) as 32-bit values at every iteration. For a standard setup with 15 iterations, this results in **960 bits/neuron** (32 bits $\times$ 2 variables $\times$ 15 iterations).
> *   **DiffPC:** Relies on event-driven ternary spikes.
>
> In our experiments (Added to the experiments section of the paper), DiffPC achieves **similar accuracy** to the floating-point baseline while transmitting only **1.9 bits/neuron** on average. This represents a reduction in data movement of over two orders of magnitude, directly illustrating the implementation efficiency gains of our method.
>
> > **The paper lacks a conclusion/perspective section, is it on purpose?**
>
> This was an unfortunate omission due to space constraints in the initial submission. We agree that a conclusion is important for framing the work. We have added a **Conclusions** section to the revised manuscript which now fits within the one extra page we are allowed to expand our work with.
>
> > **Table 4 and related content could be improved... the mean absolute difference is not very meaningful.**
>
> After reviewing the underlying assumptions and objectives of the analysis, we believe that the Mean Absolute Difference (MAD) is an appropriate metric for the type of validation shown in Table 4. We would welcome the reviewer’s thoughts on this reasoning and/or alternative suggestions.
>
> This experiment was designed to isolate the numerical precision of the DiffPC update rule from task-specific dynamics. By using random input data and shared random weights across both the standard PC and DiffPC networks, we effectively probe the approximation error across a general state space. Because the weights and inputs are drawn from standard distributions (e.g., inputs $\sim U[-1, 1]$ and Kaiming initialization), the signal magnitudes are implicitly normalized. In this context, the absolute difference values are directly interpretable as relative error. We believe this offers a rigorous, task-independent confirmation that the DiffPC formulation mathematically converges to the standard PC dynamics as the quantization parameters are tightened.
>
> > **Although some data on energy efficiency are provided, it is not clear of they translates to real world implementation efficiency.**
>
> We agree with the reviewer that simulation metrics do not capture the full complexity of real-world implementation efficiency. As we are validating our approach via simulation rather than physical measurements, we cannot accurately report calibrated energy numbers (e.g., Joules per inference). Consequently, we have revised the text to be more precise: we now frame our efficiency claims strictly in terms of **Communication Sparsity** and **Data Movement Reduction**.

---

### Official Review · Reviewer_8biP · 2025-11-09

**Soundness:** 2
**Presentation:** 1
**Contribution:** 2
**Rating:** 4
**Confidence:** 3

**Summary:**

The manuscript proposes Difference Predictive Coding (DiffPC)—a spike-native reformulation of predictive coding (PC) for spiking neural networks (SNNs). DiffPC replaces dense floating-point message passing with sparse ternary spikes for both state and error communication, introduces adaptive threshold schedules to drive event-based updates, and targets on-chip feasibility by aligning operations with Intel Loihi 2’s instruction model (validated in simulator). On MNIST and Fashion-MNIST with MLP backbones, DiffPC reports up to 99.3% and 89.6% test accuracy, respectively, while dramatically cutting communicated bits per neuron during training (e.g., 0.18 bits across 120 timesteps for DiffPC-L versus 32–960 bits for backprop/PC baselines). The method is positioned as a hardware-aligned, energy-efficient alternative to backpropagation and prior PC-SNN hybrids.

**Strengths:**

1. Spike-native PC formulation. Clear conceptual shift from continuous residuals to ternary spike streams for both targets and errors, enabling fully spiking training and inference under local rules.

2. Hardware awareness. The algorithm and simulator validation target Loihi 2 semantics, improving credibility for on-chip training pathways.

**Weaknesses:**

1. Evaluation scope (datasets & models). Experiments are restricted to MNIST/Fashion-MNIST and MLP backbones. The claims of hardware-aligned efficiency would be more compelling on spatiotemporal or event-camera benchmarks, and with convolutional or recurrent SNNs typical of neuromorphic use.

2. Learning rule completeness (weights). The PC weight update (Eq. 6) is reviewed, but Algorithm 1 does not explicitly show the weight-update pathway in the spike-only regime (e.g., how ternary error/state spikes translate into synaptic updates compatible with Loihi). This leaves ambiguity about exact stochasticity/quantization and memory consistency.

3. Inference protocol and latency. Inference appears iterative/event-driven; it would help to quantify latency–accuracy trade-offs (how many timesteps to reach stable predictions) and to compare against rate-coded SNNs with fixed horizons.

4. Energy model granularity. Bits-per-neuron is informative, but ignores NoC hops, memory hierarchies, spike routing costs, and neuron state updates on specific hardware; without this, absolute energy claims remain provisional.

**Questions:**

1. Exact synaptic update mechanics. How are weight updates computed and applied in the purely spiking formulation at run time? Is Eq. (6) implemented via accumulated spike counts/low-precision accumulators, and how are precision/overflow handled on Loihi 2? Please make this explicit in Algorithm 1 (or add an Algorithm 2 for weight updates).

2. Scheduler sensitivity & autotuning. Can you provide ablations on 𝑛, 𝑎, 𝑇, 𝑐 showing accuracy, spikes/neuron, and timesteps across datasets, and discuss any rule-of-thumb for choosing them at larger scales? Table 4 addresses approximation error only.

3. Hardware-level metrics. Do you have on-chip measurements (energy per inference/training iteration, latency, utilization) or realistic cycle-accurate estimates on Loihi 2 for DiffPC vs. PC-SE and surrogate-gradient SNNs?

---

> ### Author Response · Authors · 2025-11-21
>
> We sincerely thank the reviewer for their insightful and detailed assessment of our manuscript. We particularly appreciate the critique regarding the experimental scope and the need for greater clarity regarding the synaptic update mechanisms. We have carefully considered these criticisms and believe that addressing them has significantly strengthened the paper's contribution. Below, we detail the specific revisions and new experiments conducted in response to the reviewers comments.
>
> ### Responses to Specific Points
>
> > **Evaluation scope (datasets & models). Experiments are restricted to MNIST/Fashion-MNIST and MLP backbones. The claims of hardware-aligned efficiency would be more compelling with convolutional or recurrent SNNs typical of neuromorphic use.**
>
> We agree with the critique regarding the scope of our evaluation. While our initial reliance on MLP backbones was intended to align with prior work, we agree that demonstrating DiffPC’s viability requires validation on more complex tasks. Accordingly, we have significantly expanded our experimental section by implementing a convolutional variant of DiffPC (architecture: 10C5S2-5C5S2-50FC-30FC-10FC) and evaluating it on the **CIFAR-10** dataset.
>
> *   **Results:** The convolutional DiffPC network achieves a test accuracy of **65.6%**, which notably outperforms a standard backpropagation baseline (63.5%) using the exact same architecture and depth (Table 4 in revised paper).
> *   **Efficiency:** Crucially, this performance is achieved while maintaining high communication sparsity (approx. 1.9 bits/neuron).
>
> These new results, included in the revised manuscript, confirm that the DiffPC learning rule extends to convolutional networks without requiring fundamental changes to the algorithm.
>
> > **Exact synaptic update mechanics. How are weight updates computed and applied in the purely spiking formulation at run time? Is Eq. (6) implemented via accumulated spike counts/low-precision accumulators, and how are precision/overflow handled on Loihi 2? Please make this explicit in Algorithm 1.**
>
> Thank you for pointing this out. Our paper describes the state and error dynamics of DiffPC in detail, but the synaptic update pathway was not presented explicitly in the same level of operational detail, which may have caused confusion. We apologize for this lack of clarity.
>
> Synaptic updates do not occur on every spike. Instead, they occur at the end of the settling phase using the locally accumulated states. The update rule for a single sample is computed as:
>
> $$
> \Delta W_{ij} \propto - \left( e_{T,i} \cdot \phi(x_{T,j}) \right)
> $$
>
> where $e_{T,i}$ is the post-synaptic error state accumulated from error spikes, and $x_{T,j}$ is the pre-synaptic activity state accumulated from activity spikes. (This was explicitly added to Algorithm 1, and also to appendix section 6.2)
>
> **Regarding hardware implementation and precision:** This computation aligns directly with the programmable microcode of the **Loihi 2 learning engine**, which supports fixed-precision multiplication (e.g., `MUL_SHR`) and basic arithmetic on local variables. The architecture supports mixed-precision learning, allowing state variables to be maintained in higher precision (up to 32-bit) to capture fine-grained updates before applying them to lower-precision weights.
>
> **Regarding overflow:** The risk of accumulator overflow is structurally mitigated by our network design. First, we can employ clipped activation functions (specifically ReLU6), which impose a strict, hard upper bound on the pre-synaptic term $\phi(x_T)$. Second, the post-synaptic error term $e_T$ is derived from the forward prediction $x_F$. Since $x_F$ is a linear combination of bounded inputs scaled by synaptic weights, and the weights themselves are constrained by **weight decay**, the magnitude of the error term is structurally prevented from growing arbitrarily large. This provides a strong guarantee that the update product remains within the safety margins of standard high-precision accumulators, with hardware-level saturation available as a final safeguard.

---

> ### Author Response · Authors · 2025-11-21
>
> > **Scheduler sensitivity & autotuning. Can you provide ablations on $n, a, T, c$ showing accuracy, spikes/neuron, and timesteps across datasets, and discuss any rule-of-thumb for choosing them?**
>
> This is a highly appreciated suggestion. We have performed a grid-search ablation study on the scheduler parameters and added a **Hyperparameter Analysis** section to the Appendix (Section 6.4) and **Hyperparameter Selection Heuristics** section (Section 6.5).
>
> *   **Cycle Length ($n$):** Our analysis shows a direct trade-off between simulation latency and precision. Increasing $n$ improves the quantization resolution, leading to higher accuracy, but linearly increases the number of timesteps required.
> *   **Decay Factor ($a$):** We found that the decay factor $a$ is critical when $n$ is small (low latency). In this regime, tuning $a$ allows the threshold to decay more aggressively, recovering precision that would otherwise require a larger $n$.
>
> Regarding the other parameters: We fixed the parameter $T$ (phase length) because it governs the convergence of the underlying predictive coding equilibrium—a behavior well-characterized in standard PC literature—rather than the novel spiking quantization introduced by DiffPC. Similarly, the parameter $c$ is specific to the constant-decay scheduler variant, which was not used in the primary results. Therefore, our analysis focuses on $n$ and $a$.
>
> Based on these experiments, we propose a concrete heuristic: maximize $n$ until accuracy saturates, then reduce $n$ to improve efficiency while simultaneously decreasing $a$ to maintain representational fidelity.
>
> > **Energy model granularity. Bits-per-neuron is informative, but ignores NoC hops, memory hierarchies, spike routing costs... absolute energy claims remain provisional.**
>
> We agree with the reviewer’s assessment. "Bits-per-neuron" is a useful architecture-agnostic proxy, but it does not capture the full complexity of on-chip energy consumption.
>
> As we are validating our approach via Loihi 2 simulator rather than physical measurements on a Loihi 2 chip, we cannot accurately report calibrated energy numbers (Joules/inference). We have revised the text to be more precise: we now frame our efficiency claims strictly in terms of **Communication Sparsity** and **Data Movement Reduction**. Given that data movement is often the dominant energy cost in digital accelerators, we believe this remains a strong result, but we have removed any claims that might imply we measured total system energy.
>
> > **Inference protocol and latency. Inference appears iterative/event-driven; it would help to quantify latency–accuracy trade-offs.**
>
> We have addressed this through the ablation study in Appendix section 6.4. In our framework, the cycle length parameter $n$ dictates the number of timesteps allocated for the quantization process. By varying $n$, we directly explore the latency-accuracy trade-off. Our results demonstrate that a larger $n$ (higher latency) allows for higher precision and accuracy, while a smaller $n$ reduces the timestep count significantly with only a moderate drop in performance (which can be partially mitigated by tuning the decay factor $a$).

---

> > ### Comment · Reviewer_8biP · 2025-11-27
> >
> > I would like to thank the authors for their detailed response, which has addressed most of my earlier concerns. However, I still have two major remaining concerns:
> >
> > 1. **Limited novelty.** The core idea of the paper appears to be incremental. Similar concepts have already been explored in [1] and [2], and the main contribution here seems to be applying this line of thought to the setting of local learning with SNNs.
> >
> > 2. **Questionable practical impact.** In my view, the most pressing challenges for local learning are the poor performance of existing models and the difficulty in training large-scale networks, rather than pushing “lightweight” design to the extreme. Methods that focus primarily on extreme lightweightness, while potentially interesting from an efficiency perspective, may have very limited practical value unless they can also demonstrate competitive performance and scalability.
> >
> > >[1] Latent Weights Do Not Exist: Rethinking Binarized Neural Network Optimization.\
> > >[2] BSO: Binary Spiking Online Optimization Algorithm.

---

> > > ### Author Response · Authors · 2025-11-27
> > >
> > > ### Response to Concern 1 (Limited Novelty & Citations [1, 2])
> > >
> > > We believe there is a misunderstanding regarding the scope of our work compared to the cited references.
> > >
> > > **1. Quantized Weights vs. Quantized Communication**
> > > References [1] and [2] address **Binary Neural Networks (BNNs)**, which maximize computational and storage efficiency by binarizing synaptic **weights**. DiffPC is **not** a BNN method; we utilize standard precision weights to maintain model capacity. Instead, DiffPC introduces sparsity to the **dynamic state variables** (prediction errors and activities). This addresses a distinct bottleneck crucial for neuromorphic hardware: **communication bandwidth**.
> > >
> > > **2. Novelty in Predictive Coding Dynamics**
> > > While previous PC implementations relied on dense floating-point message passing during the iterative relaxation phase due to algorithmic constraints (training), DiffPC enables this phase to function entirely in the spiking domain.
> > >
> > > Our method leverages the specific "convergence" behavior of PC's relaxation phase. As the network settles, state changes naturally diminish. Our update rules are specifically designed to exploit this: neurons only transmit spikes when accumulated changes exceed a threshold.
> > >
> > > Crucially, our **Cyclic Threshold Scheduler** complements this by lowering the threshold over time. This allows the system to capture coarse updates early on and fine-grained updates as the network settles.
> > >
> > > ### Response to Concern 2 (Practical Impact & Scalability)
> > >
> > > We agree with the reviewer that scalability and high performance are the ultimate goals. However, we argue that in the context of neuromorphic implementation, communication efficiency is not merely an optimization, it is a prerequisite for scaling.
> > >
> > > **1. Efficiency as an Enabler for Scalability**
> > > The reviewer suggests a trade-off between efficiency and scalability. We argue that for neuromorphic hardware, they are inextricably linked. Standard PC relies on bidirectional dense message passing (errors backward, predictions forward). On neuromorphic chips (e.g., Loihi 2), which rely on local mesh-based interconnects, this dense traffic creates a massive communication bottleneck that prevents scaling. By reducing data movement by over two orders of magnitude, DiffPC transforms PC from an algorithm that saturates the bandwidth of a small network into one that can fit within the constraints of a large-scale neuromorphic system.
> > >
> > > **2. Orthogonality to Algorithmic Scaling**
> > > DiffPC is designed as a spike-based communication protocol for Predictive Coding. It does not alter the underlying objective function or the inference mechanism; it only changes the representation and transmission of messages. Recent work (e.g., *Innocenti et al., "$\mu$-PC: Scaling Predictive Coding to 100+ Layer Networks", 2025*) demonstrates that PC can scale to deep ResNets when inference dynamics are stabilized.
> > >
> > > Since DiffPC constitutes a faithful discretization of the underlying PC dynamics, it is reasonable to hypothesize that these stabilization techniques transfer directly. Naturally, this requires empirical validation. A complete scaling study would examine: (i) the average layer-wise deviation between DiffPC and continuous PC states as a function of depth; and (ii) the emergence of iso-error curves, i.e., whether deeper networks require longer spike-based communication windows to maintain approximation fidelity. These analyses constitute a dedicated study of discretized dynamics which we will highlight as future work in the conclusion section of the paper.
> > >
> > > To conclude, we highlight a brief counterfactual: had the broader research community already resolved the scaling issues of Predictive Coding, rendering it superior to Backpropagation today, a method that reduces its communication overhead by two orders of magnitude would likely be viewed as a critical contribution.

---

> > > > ### Comment · Reviewer_8biP · 2025-11-28
> > > >
> > > > I have fully understood the author's response. I had some misunderstanding regarding the innovation aspect earlier. While I remain skeptical about the scalability of this method, it cannot be denied that it offers a potential direction for the future development of local learning. Therefore, I have decided to raise the score from 4 to 6.

---

> > > > > ### Author Response · Authors · 2025-11-28
> > > > >
> > > > > We sincerely thank the reviewer for the engaging discussion and for re-evaluating our work. We appreciate your open-mindedness regarding the novelty of DiffPC, and acknowledge your points on scalability; these concerns will definitely guide our future work as we move from efficient architectural designs to large-scale implementations. We are very grateful for your decision to raise the score to 6. We noticed that the official score field on the dashboard has not yet been updated to reflect this change, so we kindly ask you to update it at your convenience.

---

### Author Response · Authors · 2025-12-02

We thank all reviewers for their insightful feedback and the engaging discussions, which helped us improve the manuscript.

**For the Area Chair**

Due to the reverting of review scores and the suspension of the discussion period, we provide this brief summary to assist in your evaluation:

*   **Reviewer 8biP (Score 6):** Stated in the discussion that they raised their score to 6 following our clarification on novelty and the addition of CIFAR-10 experiments.
*   **Reviewer 8Q9N (Score 4):** Raised their score to 4 after reviewing our new experimental results. We were still discussing their remaining concerns when the discussion period ended.
*   **Reviewer VdSN (Score 4):** Did not have time to respond to our rebuttal. However, we implemented all their requested changes, including the missing scheduler definitions, hyperparameter analysis, and efficiency comparisons.
*   **Reviewer htVv (Score 4):** We implemented their requested changes, which they acknowledged as satisfactory early in the rebuttal. We subsequently engaged in a detailed discussion regarding the theoretical differences between DiffPC and Backpropagation. We addressed their final request by providing specific literature comparisons on the advantages of PC, but the discussion period ended before they could update their score.

**Key Concerns Addressed**

A key concern raised by multiple reviewers was the limited experimental scope (MLP networks). In response, we implemented a Convolutional DiffPC variant and evaluated it on CIFAR-10, demonstrating that it outperforms the Backpropagation baseline while maintaining high communication sparsity.

Regarding the critique that we did not demonstrate scaling to deep networks, we provided context that this is a limitation of the standard Predictive Coding (PC) framework itself, which struggles with depth, rather than a limitation of our spiking formulation. We highlighted very recent emerging work (e.g., $\mu$-PC, 2025) which demonstrates that PC can scale to 100+ layers when specific stabilization techniques are applied to the inference dynamics.

We noted that the convolutional architectures evaluated in our work are already approaching the depth scaling limit of standard, unstabilized PC. Consequently, we argued that while emerging stabilization techniques can likely be utilized to scale DiffPC, identifying which technique is the optimal fit and verifying the integration requires a dedicated future study.

Finally, we emphasized that while PC offers distinct theoretical advantages over Backpropagation (e.g., reduced catastrophic forgetting in online learning, improved sample efficiency), standard PC incurs significantly higher compute and data transfer costs. DiffPC attains more than 500x reduction of data transfer compared to standard PC and thus takes PC one important step forward to be a true alternative to BP.

---

### Meta-Review · Area_Chair_Fj3w · 2026-01-06

**Summary:**

This paper proposes Difference Predictive Coding (DiffPC), a spike-native reformulation of predictive coding that replaces dense floating-point communication with sparse ternary spike-based message passing. Reviewers broadly agree that the work is technically sound and represents a meaningful step toward making predictive coding compatible with neuromorphic hardware, particularly through its focus on communication sparsity and local learning. While the initial submission suffered from limited experimental scope and presentation issues, the revised version substantially improves clarity and adds convolutional experiments on CIFAR-10, strengthening the paper’s positioning as a hardware-aligned learning framework rather than a purely theoretical contribution.

**Reviewer Concerns:**

Most major concerns raised by reviewers were addressed in the rebuttal and revision.

**Reviewer Scores:**

Reviewer scores show a clear upward trend after the rebuttal.

---

### Decision · Program_Chairs · 2026-01-26

Accept (Poster)